EMBO
Molecular Medicine

# Diabetes drugs activate neuroprotective pathways in models of neonatal hypoxic-ischemic encephalopathy

Laura Poupon-Bejuit[1,4], Amy Geard [ID][1,4], Nathan Millicheap[1], Eridan Rocha-Ferreira [ID][2], Henrik Hagberg [ID][2], Claire Thornton [ID][3] & Ahad A Rahim [ID][1✉]

## Abstract

Hypoxic-ischaemic encephalopathy (HIE) arises from diminished blood flow and oxygen to the neonatal brain during labor, leading to infant mortality or severe brain damage, with a global incidence of 1.5 per 1000 live births. Glucagon-like Peptide 1 Receptor (GLP1-R) agonists, used in type 2 diabetes treatment, exhibit neuroprotective effects in various brain injury models, including HIE. In this study, we observed enhanced neurological outcomes in post-natal day 10 mice with surgically induced hypoxic-ischemic (HI) brain injury after immediate systemic administration of exendin-4 or semaglutide. Short- and long-term assessments revealed improved neuropathology, survival rates, and locomotor function. We explored the mechanisms by which GLP1-R agonists trigger neuroprotection and reduce inflammation following oxygen-glucose deprivation and HI in neonatal mice, highlighting the upregulation of the PI3/AKT signalling pathway and increased cAMP levels. These findings shed light on the neuroprotective and anti-inflammatory effects of GLP1-R agonists in HIE, potentially extending to other neurological conditions, supporting their potential clinical use in treating infants with HIE.

**Keywords** GLP1-R Agonists; Semaglutide; Exendin-4; Neonatal Hypoxic-ischaemic Encephalopathy; Neuroprotective Mechanisms
**Subject Category** Neuroscience

## Introduction

Hypoxic-ischaemic encephalopathy (HIE) is one of the leading causes of brain injury in infants, with a high risk of mortality and disability (Volpe, 2012). It affects 1–3 per 1000 live births in developed countries, and 26 per 1000 in the developing world (Kurinczuk et al, 2010). An inadequate oxygen supply and blood flow results in a variety of clinical manifestations (Allen and Brandon, 2011; Ferriero, 2004), including developmental delays, epilepsy, seizures, cerebral palsy, and death

(Dilenge et al, 2001; Shankaran, 2012). Of the affected neonates, 15–25% die during the neonatal period and 25% of the survivors develop neurological sequelae (Lai and Yang, 2011). The pathophysiology of hypoxia-ischaemia (HI) brain damage involves inflammation, oxidative stress, excitotoxicity and cell death (Gluckman and Williams, 1992; Lorek et al, 1994; Penrice et al, 1996; Rocha-Ferreira and Hristova, 2016). Two different phases of neuronal death have been identified in both clinical and experimental studies (Dixon et al, 2015; Penrice et al, 1996). First, exhaustion of the cell's energy stores induces immediate neuronal death and, second, delayed neuronal death occurs after a latent period of up to 6 h, which is associated with encephalopathy present in the latent phase and increased seizure activity. The current treatment is therapeutic hypothermia (TH), in which a reduction in either head or body temperature reduces long-term effects when applied in moderate to severe injury (Davidson et al, 2015). However, in most cases, hypothermia is not a sufficient treatment as 50% of infants with moderate to severe HIE die or survive with disability. The benefit is greater in moderate HIE (Edwards et al, 2010). Thus, further studies on improving TH success rates or finding therapeutic alternatives are urgently required.

A number of studies have shown that peptide agonists of the glucagon-like peptide 1 receptor (GLP1-R), which are licensed for the treatment of type 2 diabetes mellitus (T2DM), also have potent neuroprotective and anti-inflammatory properties (Liu et al, 2015; Zhang et al, 2019). A clinical trial in Parkinson's disease patients using systemic administrations of the GLP1-R agonist exendin-4 in an extended release formulation has demonstrated clinical efficacy (Athauda et al, 2017). We have previously reported that exendin-4 also has significant neuroprotective and anti-inflammatory properties when systemically administered either immediately after HI induced by unilateral carotid occlusion in day 7 postnatal CD1 mice (P7), or 4 h after HI (Rocha-Ferreira et al, 2018). This therapeutic efficacy was enhanced through combined synergistic treatment with hypothermia (Rocha-Ferreira et al, 2018). However, the mechanism behind the therapeutic effect of exendin-4 following HI was not investigated. Furthermore, there has been development of GLP1-R peptide agonists allowing a significant extension in the half-life of the drug in circulation. An example of this is semaglutide (Ozempic®) that has been

[1]Department of Pharmacology, UCL School of Pharmacy, University College London, London WC1N 1AX, UK. [2]Institute of Clinical Sciences, Sahlgrenska Academy, University of Gothenburg, Gothenburg, Sweden. [3]Department of Comparative Biomedical Sciences, Royal Veterinary College, London, UK. [4]These authors contributed equally: Laura Poupon-Bejuit, Amy Geard. ✉E-mail: a.rahim@ucl.ac.uk

approved in the USA and Europe as a once-weekly treatment for T2DM (Dhillon, 2018; Hedrington et al, 2018). Semaglutide has a half-life of 7 days compared with 60-90 min for exendin-4 (Nielsen et al, 2004). Preclinical studies have already demonstrated the neuroprotective effects of semaglutide in murine models of stroke (Basalay et al, 2019; Yang et al, 2019) and Parkinson's disease (Zhang et al, 2018; Zhang et al, 2019). A phase II clinical trial testing semaglutide in patients with Parkinson's disease is also in progress (NCT03659682).

In this study, we investigated the underlying cellular mechanisms to decipher how GLP1-R agonists exendin-4 and semaglutide protect the brain in vitro using primary neurons exposed to oxygen-glucose deprivation (OGD) to induce HI, and in vivo where HI was induced by unilateral carotid occlusion in P10 CD1 mice, and either saline, exendin-4 or semaglutide were given immediately after HI. We show that key neuroprotective pathways are upregulated while pro-cell death pathways are downregulated, and neuroinflammatory and oxidative molecules are modulated. We also conducted in vitro and in vivo studies to directly compare the therapeutic efficacy of a single dose of longer-acting semaglutide versus multiple doses of shorter acting exendin-4. Both drugs are highly effective and provide short- and long-term therapeutic efficacy when assessed in P10 CD1 mice following induced HI. Therapeutic efficacy was observed when investigating survival, neuropathology and behavioural markers, although blood glucose levels required supplementation when using high doses of semaglutide in neonatal animals that received HI. This study sheds light onto the mechanisms by which GLP1-R protects the brain, which may be beneficial for other neurological conditions. It also provides further evidence supporting the potential clinical translation of GLP1-R agonists for the treatment of neonatal HIE.

# Results

## Semaglutide and exendin-4 activate GLP1-R in vitro

To compare the ability of exendin-4 and semaglutide to activate GLP1-R, we used a previously published GLP1-R/pCRE-luciferase/CHO-K1 cell line suitable for monitoring the activity of GLP1-R through luminescence readout (Garry et al, 2015; Kim et al, 2009). Cells were treated with 1 μM exendin-4 or semaglutide, with or without GLP1-R antagonist, exendin-9 (Ex-9) and subsequent GLP1-R activation was assessed by luciferase production 2 h after treatment (Fig. 1A). A statistically significant increase in GLP1-R activation was observed using 1 μM of either semaglutide or exendin-4 compared to untreated cells ($p < 0.0001$ and $p = 0.0009$, respectively). This luciferase expression was inhibited following treatment with exendin-9 (Fig. 1A). GLP1-R activation was also confirmed through intracellular measurements of the secondary messenger, cyclic adenosine monophosphate (cAMP), in the GLP1R/pCRE-luciferase/CHO-K1 cells (Fig. 1B). Compared with untreated cells, the cAMP content was significantly increased in the cell lysate and medium of cells treated with 1 μM of semaglutide ($p < 0.0001$ and $p < 0.0001$, respectively) or 1 μM exendin-4 ($p = 0.0004$ and $p < 0.0001$, respectively). Images of luciferase bioluminescence was also captured using an in vivo imaging system (IVIS) (Fig. EV2D).

## GLP1-R activation reduces oxidative stress and cell death in primary neuronal cultures exposed to oxygen-glucose deprivation (OGD)

Using in vitro primary neuronal cultures that have undergone OGD to induce ischaemic injury, we explored whether exendin-4 or semaglutide could be a neuroprotective agent, as well as their impact on oxidative stress and cell death. By using this in vitro primary neuronal cell approach, we were able to focus on neurons and mechanisms involved in neuroprotection without other neural cells to obfuscate the readouts. OGD leads to stabilised Hypoxia-inducible factor (HIF)1-α expression and stress-induced ATF3 expression. In response to GLP1-R agonists, GLP1-R initiates cAMP-mediated and/or phosphoinositol-3 kinase (PI3K)-mediated signalling potentially limiting OGD-induced cell death pathways (e.g. via GSK3β) or increasing expression of anti-apoptotic proteins (e.g. Bcl-xL; Fig. 2A) (Holscher, 2014).

HIF-1α is a transcription factor that responds to changes in oxygen levels and has been found to be upregulated in brains after hypoxic and ischaemic exposures (Shi, 2009). Stabilised HIF-1-α can induce a variety of responses to hypoxia ranging from prosurvival to prodeath depending on the severity of the insult (Piret et al, 2002); in the neonatal brain, HIF-1-α has been shown to exert protective effects in neonatal HI (Sheldon et al, 2009). Therefore, we examined the levels of HIF-1-α after OGD with or without GLP1-R agonist treatment. A significant increase in HIF-1-α expression was measured by qPCR in the OGD model compared to control cells ($p = 0.0006$), but this increase was statistically significantly reduced and normalised when neurons were treated with exendin-4 or semaglutide ($p < 0.0001$ and $p < 0.0001$, respectively) (Fig. 2B). The addition of Ex-9 inhibited this improvement, confirming that the effect of both exendin-4 and semaglutide are specifically mediated through activation of GLP1-R.

The activation of transcription factor 3 (ATF3) is observed in various tissues in response to stress and is used as a marker of neuronal damage (Hunt et al, 2012; Tsujino et al, 2000). In neurons exposed to OGD, a significant increase of ATF-3 ($p < 0.0001$) was detected by quantitative PCR (qPCR) compared to control cells that received saline (Fig. 2C). A significant decrease in ATF3 was measured by qPCR and observed in cells by immunofluorescence following treatment with both GLP1-R agonists ($p = 0.0378$ for exendin-4, $p = 0.0226$ for semaglutide), suggesting protection against neuronal damage. The addition of Ex-9 inhibited this effect.

The cAMP response element-binding (CREB) protein mediates genes closely associated with neuronal survival, neural differentiation, and neurite outgrowth (Holz et al, 2006; Lonze et al, 2002). CREB also appears to be an important protective component in the immature brain in response to ligation preconditioning (Lee et al, 2004). Therefore, we examined the gene expression levels of CREB by qPCR in the neurons exposed to OGD. The OGD cultures that were treated with control saline showed a significant decrease in CREB compared to control cells treated with saline ($p < 0.0001$). However, GLP1-R agonists increased and normalised the expression levels of CREB in the OGD cells ($p = 0.0016$ for exendin-4, $p = 0.0002$ for semaglutide) (Fig. 2D). Again, the antagonist exendin-9 blocked this effect suggesting a specific role for activation of GLP1-R in neuroprotective mechanisms. Further neuroprotective mechanisms of GLP1-R agonists were investigated through interleukin-1 beta (IL-1β). IL-1β, the only interleukin produced in neuron cultures, is a proinflammatory interleukin shown to increase

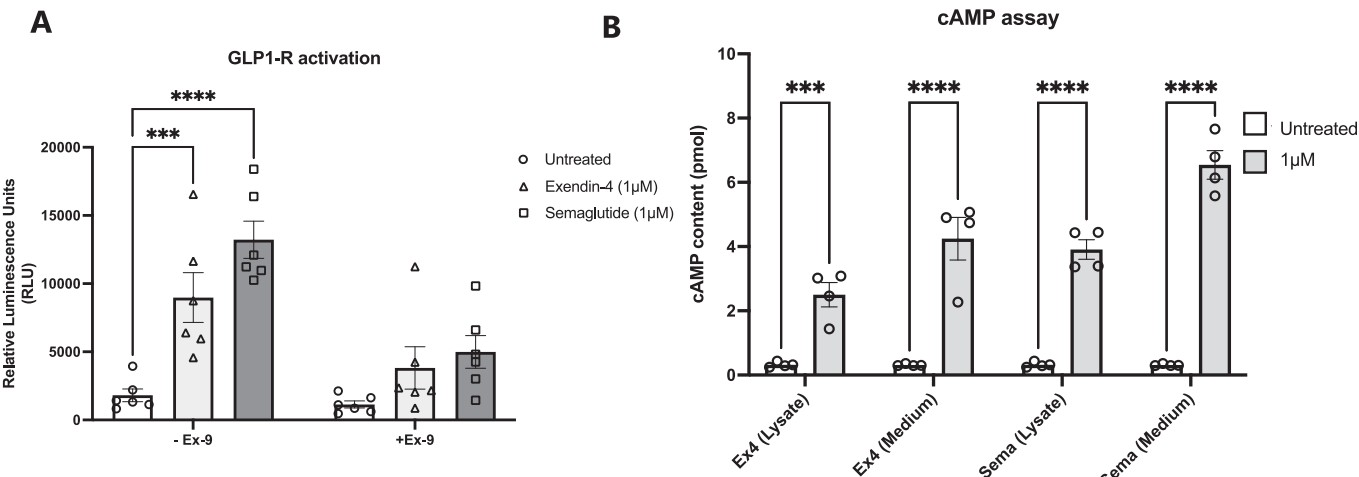

**Figure 1. Semaglutide and exendin-4 activate the GLP1 receptor in CHO-GLP1-R cells.**

GLP1R/pCRE-luciferase/CHO-K1 cells treated with 1 μM of exendin-4 or semaglutide with or without antagonist, exendin9-39 (500 nM) compared with untreated cells (n = 6 wells/group). (A) Measurement of luciferase activity induced by GLP1-R activation with exendin-4 and semaglutide. (B) cAMP levels measured in cell lysate and media after treatment with GLP1-R agonists compared with untreated cells. Data information: Error bars indicate mean ± SEM. Statistical analysis was performed using an ordinary two-way analysis of variance (ANOVA) corrected for using (A) Tukey's multiple comparisons test and (B) Sidak's multiple comparison test. ***P < 0.001, ****P < 0.0001. Source data are available online for this figure.

following neonatal HI injury (Hagberg et al, 1996; Ji et al, 2015) and to enhance injury (Hagberg et al, 1996). IL-1β expression level increased in cells exposed to OGD compared to control cells (****$p < 0.0001$), indicating that neuroinflammatory responses are activated (Fig. EV2A). However, the levels of IL-1β are significantly reduced following application of both GLP1-R agonists ($^{##}p = 0.0020$ for exendin-4 and $^{##}p = 0.0057$ for semaglutide).

Neuronal death was observed 2 h after OGD by a significant decline in survival to 67% (Fig. 2E) ($p < 0.0001$). Treatment with GLP1-R agonists significantly increased the survival of OGD-treated neurons at a dose of 1 μM ($p < 0.0001$ for both treatments) with 83.5% and 87.4% survival for exendin-4 and semaglutide, respectively. The role of GLP1-R activation in this neuroprotective mechanism was confirmed by blocking the receptor using the antagonist exendin 9-39 (Ex-9) which resulted in a significant decrease in neuronal survival ($p = 0.026$ for exendin-4 vs exendin-4 + exendin-9, $p = 0.0004$ for semaglutide vs semaglutide + exendin-9), but still significantly increased in comparison to the OGD saline-treated control ($p = 0.0310$ for exendin-4 + exendin-9; $p = 0.0320$ for semaglutide + exendin-9), suggesting a non-receptor effect.

Expression of the Bcl-2 family of proteins, which are essential mitochondrial apoptosis regulators, was investigated by qPCR in primary neuron cultures after OGD with or without GLP1-R agonist treatment. Following injury, treatment with GLP1-R agonists significantly increased the expression level of the anti-apoptotic Bcl-xL ($p = 0.0405$ for exendin-4, $p = 0.0013$ for semaglutide) and Bcl-2 ($p = 0.0003$ for exendin-4, $p = 0.004$ for semaglutide) genes (Fig. 2F,G) in comparison to untreated cell cultures exposed to OGD. In addition, a statistically significant reduction in the enzyme activity of caspase 3 ($p = 0.0214$ for exendin-4, $p = 0.0101$ for semaglutide) (Fig. 2H), caspase 8 ($p = 0.0049$ for exendin-4, $p = 0.0033$ for semaglutide; Fig. EV2B) and caspase 9 ($p < 0.0001$ for both treatments; Fig. EV2C) was also detected following treatment

with the GLP1-R agonists. The addition of the GLP1-R antagonist exendin-9 inhibited the anti-apoptotic effects of both exendin-4 and semaglutide, suggesting the protective effect is mediated through the activation of GLP1-R.

## Semaglutide crosses the blood-brain barrier and reduces infarct size similar to exendin-4 after neonatal HI brain injury

Before an in vivo evaluation of the mechanisms of neuroprotection and anti-neuroinflammation could be conducted, we needed to establish whether semaglutide could cross the blood-brain barrier (BBB) and activate GLP1-R following intraperitoneal (IP) administration. We confirmed that the IP route of administration allows semaglutide to cross the blood-brain barrier in naïve neonatal mice. A significant increase in cAMP levels in the brain was observed 30 min following semaglutide injection with similar efficacy as exendin-4 (**$p = 0.0091$ at 0.5 μg/g and **$p = 0.0015$ at 2 μg/g for exendin-4; ****$p < 0.0001$ all doses for semaglutide) (Fig. 3A). Entry into the brain and subsequent activation of GLP1-R was more rapid when the agonist was given IP compared with subcutaneously (SC), the usual route of administration for patients with diabetes, at 30 mins (****$p < 0.0001$ IP vs SC) and 2 h (**$p = 0.0024$ IP vs SC) (Fig. 3B).

We then investigated whether semaglutide could ameliorate damage in the P10 CD1 mice with surgically induced HI, similar to that which we previously demonstrated using exendin-4 (Rocha-Ferreira et al, 2018). An effective dosing regimen for exendin-4 was determined in our previous study (Rocha-Ferreira et al, 2018) at 0.5 μg/g given every 12 h over a 48-h period and which also did not cause hypoglycaemia. We investigated glucose levels in P10 neonatal mice that received HI before and after administration with 0.5 μg/g semaglutide at various time points and also via IP vs SC routes of administration (Fig. EV2E). Statistically significant

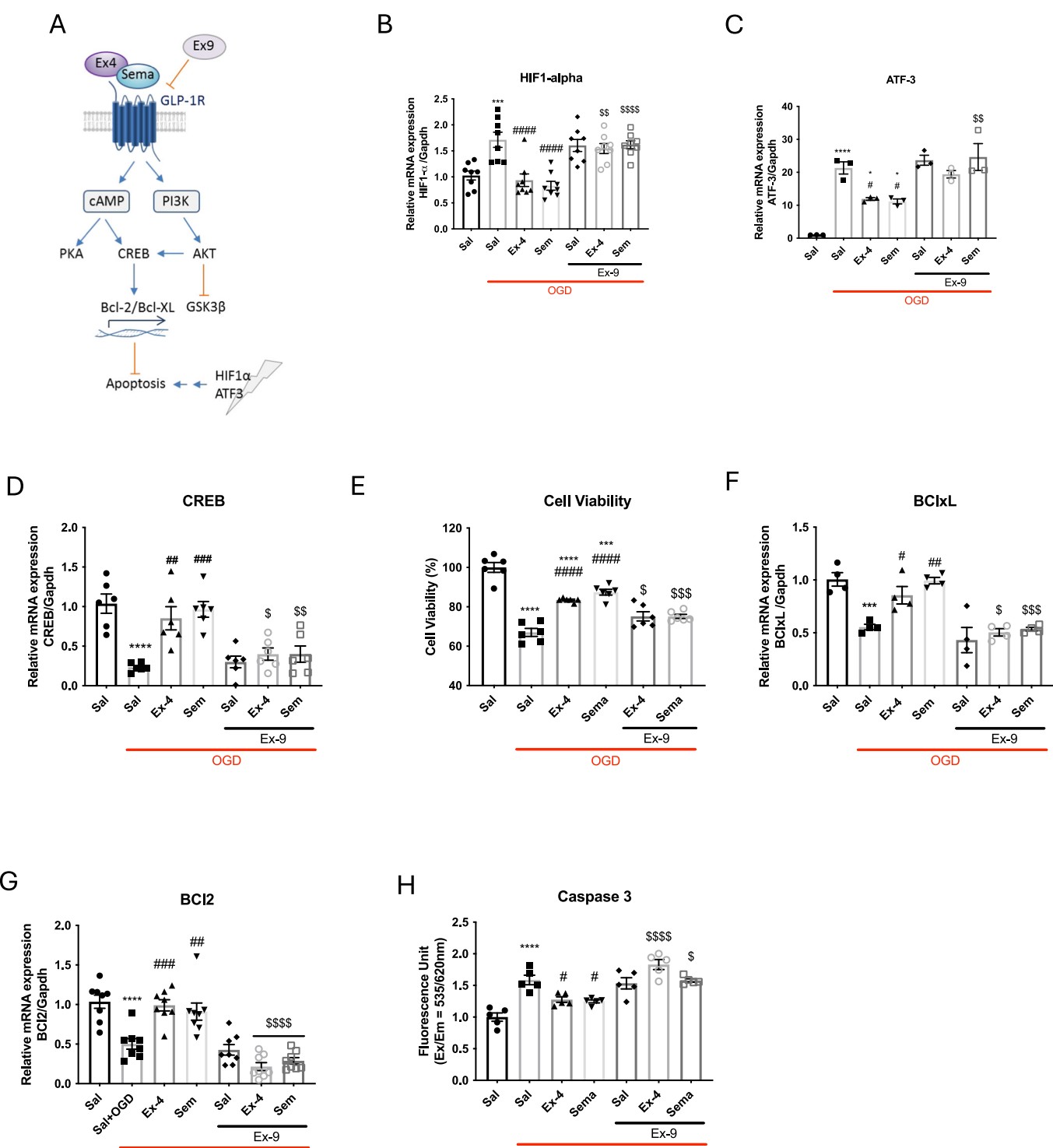

hypoglycaemia was detected at 30 min (***$p = 0.0002$ for IP), 2 h (***$p = 0.0003$ for IP, **$p = 0.0180$ for SC) and 24 h (***$p = 0.0003$ for IP, *$p = 0.0289$ for SC) post-injection with semaglutide via the indicated routes. We also observed increased mortality. However, this was completely compensated for by oral supplementation with 2 mg/g glucose (Fig. EV2E). Thereafter, a dosing experiment was conducted to evaluate the optimal dose for semaglutide in

the same neonatal HIE model (Fig. EV2F) and brain protection was measured using a macroscopic scoring system to assess the size of the infarct. Here, the addition of oral glucose supplementation prevented the increased mortality observed in neonatal mice administered with the peptide following HI, and we observed that a single dose of semaglutide (0.25 μg/g) shows the same therapeutic effect as 4 doses of exendin-4 (0.5 μg/g; Fig. EV2F).

**Figure 2. Semaglutide provides similar protection as exendin-4 following OGD in vitro.**

(A) GLP1-R agonists Exendin-4 and Semaglutide trigger GLP1-R signalling via cAMP and PI3K. Transcription of prosurvival genes and inhibition of apoptotic pathways may counteract the effect of hypoxia-mediated apoptosis. Incubation with GLP1-R antagonist Exendin 9 prevents GLP1-R activation. (B) Evaluation of neuroinflammatory marker HIF-1-α using qPCR in the in vitro neuronal cells exposed to OGD following treatment with exendin-4 or semaglutide, and in combination with exendin-9 (Ex-9) ($n = 8$ for all experimental groups). (C) Neuronal damage marker ATF-3 assessed using qPCR ($n = 3$ for all experimental groups). (D) Levels of the neuroprotective marker CREB were investigated using qPCR ($n = 6$ for all experimental groups). (E) Cell viability assessed after OGD and treatments with exendin-4, semaglutide, and combination with exendin-9 ($n = 6$ for all experimental groups). Evaluation of apoptosis in primary neuronal cell cultures exposed to OGD with measure of gene expression for (F) Bcl-xL ($n = 4$ for all experimental groups) and (G) Bcl-2 ($n = 8$ for all experimental groups). (H) Caspase 3 activity ($n = 5$ for all experimental groups). Data information: Error bars indicate mean ± SEM. Statistical analysis was performed using an ordinary one-way analysis of variance (ANOVA) corrected for using Tukey's multiple comparisons test. * or # or $, $p < 0.05$; ** or ## or $$, $p < 0.01$; *** or ### or $$$, $p < 0.001$; **** or #### or $$$$, $p < 0.0001$. * to compare with control (Sal) and # to compare with OGD group (Sal + OGD). $ symbol used to compare GLP1-R agonists treatment with corresponding treatment in combination with exendin-9. Source data are available online for this figure.

## GLP1-R agonist ameliorates short- and long-term damage to the brain and improves survival following HI injury

The well-characterised Vannucci model of neonatal brain injury was used to test whether semaglutide could protect the brain from hypoxia-ischaemia-induced damage. The short-term efficacy of a single dose of semaglutide (0.25 µg/g, supplemented with glucose) was compared with 4 doses of exendin-4 (0.5 µg/g) and the tissue infarct was measured 7 days post-insult. Significant tissue loss was observed in the ipsilateral hemisphere of saline-treated HI controls (Fig. 4A) that correlated with the macroscopic score (1.5, ***$p = 0.0001$) (Fig. 4B). Treatment with 4 doses of exendin-4 or a single intraperitoneal administration of semaglutide starting immediately after HI reduced tissue infarct volume and correlated with a reduced macroscopic score (0.35, ##$p = 0.0021$ for exendin-4; 0.26, ###$p = 0.0010$ for semaglutide) (Fig. 4B). No differences between male and female response to the treatments were noted and is consistent with our previous findings using exendin-4 (Rocha-Ferreira et al, 2018). We have previously reported on the short-term benefits of exendin-4 in P7 CD1 mice with surgically induced HI, 48 h after injury (Rocha-Ferreira et al, 2018). However, a longer-term study has never been conducted using a GLP1-R agonist in neonatal mice with induced HI. Therefore, we followed animals in each experimental group for 2 months post-injury and evaluated the neuroprotective capabilities of the drugs. The overall survival rate of the control sham saline treatment group at 60 days of age was 100%, the HI-saline group showed a reduced survival of 83.3%, whilst treatment with exendin-4 was 88.8% (Fig. 4C). However, semaglutide treatment resulted in 100% survival at 60 days. Brain architecture of the mice that survived to 2 months post-injury was examined by Nissl staining (Fig. 4D). No infarct was observed in the control sham group (saline), whereas the HI-saline group had a significant infarct area. This was supported by the macroscopic scoring (MS: 1.67, ****$p < 0.0001$) (Fig. 4E). Treatment with exendin-4 and semaglutide significantly reduced the infarct area (Fig. 4D) as was confirmed by the macroscopic score (MS: 0.41, ###$p = 0.0004$; MS: 0.45, ###$p = 0.0007$, respectively) (Fig. 4E).

## GLP1-R agonists ameliorate short-term HI-induced cellular-mediated neuroinflammation in vivo

The effect of both semaglutide and exendin-4 on microglial- and astroglial-mediated inflammatory responses was assessed in the cerebral cortex and hippocampus by immunohistochemistry and quantitative threshold image analysis. In the short-term study, microglia-specific markers CD68 (Fig. 5A), Iba-1 (Fig. 5B) and astrocytic marker GFAP (Fig. 5C) showed a significant inflammatory response in both areas examined from saline-treated HI mice compared to no HI saline controls (****$p < 0.0001$). The microglial-mediated (CD68 and Iba-1) inflammatory responses are significantly reduced with either GLP1-R agonist treatment (####$p < 0.0001$) with no significant difference compared with saline (no HI) controls in both regions of the brain. Astrogliosis is significantly reduced by exendin-4 in the cortex (cortex: #$p = 0.0134$; hippocampus: $p = 0.0604$) and prevented by semaglutide compared to control saline-treated HI mice (cortex ####$p < 0.0001$; hippocampus ###$p = 0.0010$) with no significant difference from control (no HI) saline-treated mice.

## GLP1-R agonists ameliorate longer-term HI-induced cellular-mediated neuroinflammation in vivo

We also assessed the neuroinflammatory response in the brains of long-term mice that survived to 2 months post-HI. Neuroinflammation was present in the HI saline-treated group, but this was predominantly restricted to the hippocampus. Immunohistochemistry using antibodies against CD68 revealed no activation in the cortex and quantitative threshold analysis confirmed no statistically significant increase in staining across all experimental cohorts (Fig. 6A). A statistically significant increase in macrophage activation was observed in the hippocampus for the HI saline-treated group when compared to the no HI saline group (**$p = 0.0027$). A significant reduction in macrophage activation was observed in the exendin-4 and semaglutide groups compared to HI saline groups (##$p = 0.0059$ and #$p = 0.0204$, respectively) (Fig. 6A). Using antibodies against Iba-1, a significant increase in microglial activation was observed in the hypoxic-ischaemic group in comparison with the no HI-saline treated group in both the cortex (**$p = 0.0011$) and in the hippocampus (***$p = 0.0008$) (Fig. 6B). In the cortex, a significant reduction of microglial activation was observed for semaglutide-treated mice (##$p = 0.0061$), with exendin-4 treatment tending to reduce microglial activation ($p = 0.1497$). No statistically significant differences were observed between either treatment group, and the no HI-saline group and HI cohorts in the hippocampus (Fig. 6B). Astrocyte activation using antibodies against GFAP was not observed in the cortex of any of the treatment groups (Fig. 6C). Significant astrocytic activation was observed and measured within the hippocampus of the hypoxic-ischaemic group when compared to the control saline-treated no-HI group (Fig. 6C) (****$p < 0.0001$). A statistically significant

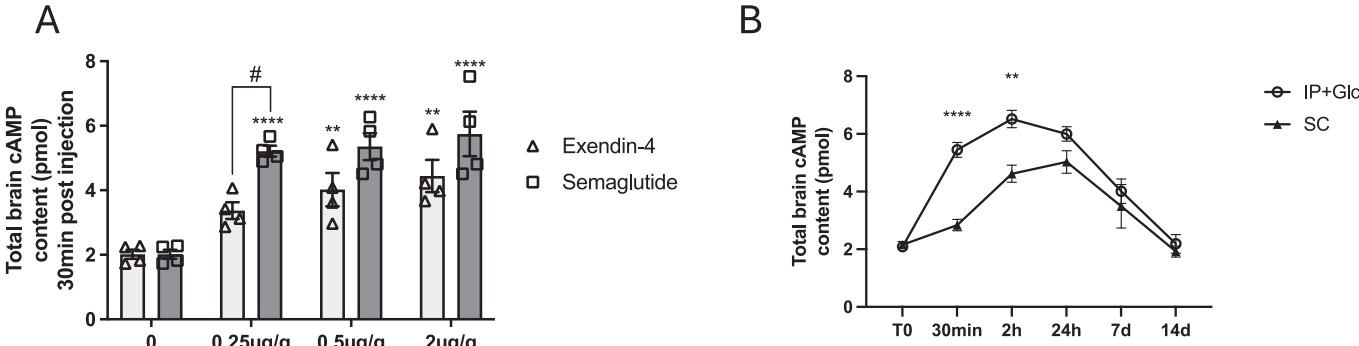

**Figure 3.  Single dose semaglutide delivered IP crosses the BBB and provides activation of the GLP1-R.**

(A) cyclic AMP (cAMP) content in the brain 30 min after IP delivery of exendin-4 and semaglutide to check their efficiency at crossing the blood-brain-barrier (BBB). * to compare to untreated mice, and # to compare between exendin-4 and semaglutide at the corresponding dose ($n = 4$ for all experimental groups). (B) Kinetics of cAMP content in the brain for validation of the IP route of administration, with glucose supplementation, and time taken to reach the brain in comparison to the standard subcutaneous route ($n = 4$ per time point for each experimental group). Data information: Error bars indicate mean ± SEM. Statistical analysis was performed using an ordinary two-way analysis of variance (ANOVA) corrected for using Sidak's multiple comparisons test. * or #, $p < 0.05$; ** or ##, $p < 0.01$; **** or ####, $p < 0.0001$. Source data are available online for this figure.

decrease in astrogliosis was observed and measured in exendin-4 and semaglutide treatment groups (####$p < 0.0001$ for both treatments). Macrophage activation was also investigated in all organs (heart, lung, liver, spleen, and kidney) and no significant activation was observed (Fig. EV4A,B).

## No adverse events associated to a single high dose administration of semaglutide in naïve P10 CD1 mice

Previously, we have reported that the four administrations of high dose exendin-4 used in this study over a 48 h period had no obvious adverse effects (Rocha-Ferreira et al, 2018). We therefore examined whether the single high dose of semaglutide used in this study had any noticeable adverse effects 24 h after intraperitoneal administration into naïve wild type P10 CD1 mice. Saline-treated mice were used as the control group. The brain, lungs, spleen, heart, liver and kidney of each mouse were harvested and examined histologically. No obvious fibrosis, or any change in tissue architecture, were observed (Fig. EV3A,B). Immunohistochemistry was conducted on tissue sections using the CD68 antibody to investigate any macrophage-mediated inflammatory response and quantified by quantitative threshold image analysis. No significant macrophage infiltration or activation was observed or measured in any organs after semaglutide treatment when compared to the saline treatment group (Fig. EV3C). Blood samples were taken, and several blood cell populations were measured, including: total white blood cells (WBCs), neutrophils, lymphocytes, monocytes, eosinophils and basophils counts, haematocrit (HCT), platelets, red blood cells (RBCs), haemoglobin and mean corpuscular volume (MCV). No significant differences were observed between semaglutide-treated groups and control saline-treated groups for all blood parameters measured (Fig. EV3D).

## Treatment with GLP1-R agonists improves behavioural outcomes with no obvious adverse effects after HI in vivo

We evaluated the locomotor function of mice at 4 weeks and 8 weeks of age in each experimental cohort using a range of tests.

The rotarod test was used to assess balance and coordination using latency to fall. Latency was significantly reduced after HI at both 4 weeks and 8 weeks in comparison to the saline (no HI) group (*$p = 0.0453$ and *$p = 0.0233$, respectively) (Fig. 7A). Exendin-4 and semaglutide treatment groups showed improvement in balance and coordination at 8 weeks of age compared with HI-saline treated mice and were comparable to control measurements.

Parameters captured following open field tests demonstrated that hypoxic-ischaemic mice showed a significant decrease in total distance travelled at 4 weeks and 8 weeks of age (****$p < 0.0001$; **$p = 0.0095$, respectively) and an increase in freezing time (****$p < 0.0001$; $p = 0.0513$, respectively), when compared to control non-hypoxic-ischaemic mice (Fig. 7B). Treatment with exendin-4 or semaglutide significantly improved these 2 parameters at 4 weeks (Distance: ####$p < 0.0001$ and ##$p = 0.0036$, respectively; Freezing Time: #$p = 0.0341$ and $p = 0.0941$, respectively) and the distance travelled at 8 weeks (Distance: $p = 0.0507$ for exendin; #$p = 0.0424$ for semaglutide), with a trend towards decreased freezing time using semaglutide.

Automated gait analysis (Noldus Catwalk XT) was used to qualitatively and quantitatively assess gait in all experimental cohorts. An examination of paw prints in control mice that didn't experience HI showed the normal pattern of hind paw prints in close proximity to the front paw prints, with a typical alternate step sequence of right front, left hind, left front and right hind (Fig. 7C). The hypoxic-ischaemic group demonstrated clear irregular gait compared to the control no HI saline group. However, treatment of hypoxic-ischaemic mice with exendin-4 or semaglutide improved gait to that comparable with control no HI saline-treated mice. Data from the CatWalk were further analysed and revealed significant exendin-4 and semaglutide induced improvements at 4 weeks and 8 weeks of age compared with the age-matched hypoxic-ischaemic group for various parameters: run duration (4W: ###$p = 0.0006$, ##$p = 0.0012$; 8W: #$p = 0.0346$, #$p = 0.0332$) average speed (4W: #$p = 0.0196$, #$p = 0.0471$; 8W: ns), regularity index (4W: #$p = 0.0147$, #$p = 0.0123$; 8W: #$p = 0.0384$, #$p = 0.0228$), stride length (4W (RH): #$p = 0.0198$, #$p = 0.0466$; 8W (LF): #$p = 0.0322$, #$p = 0.0122$) (Fig. EV4C–F at 4W; Fig. EV4G–J at 8W). To investigate

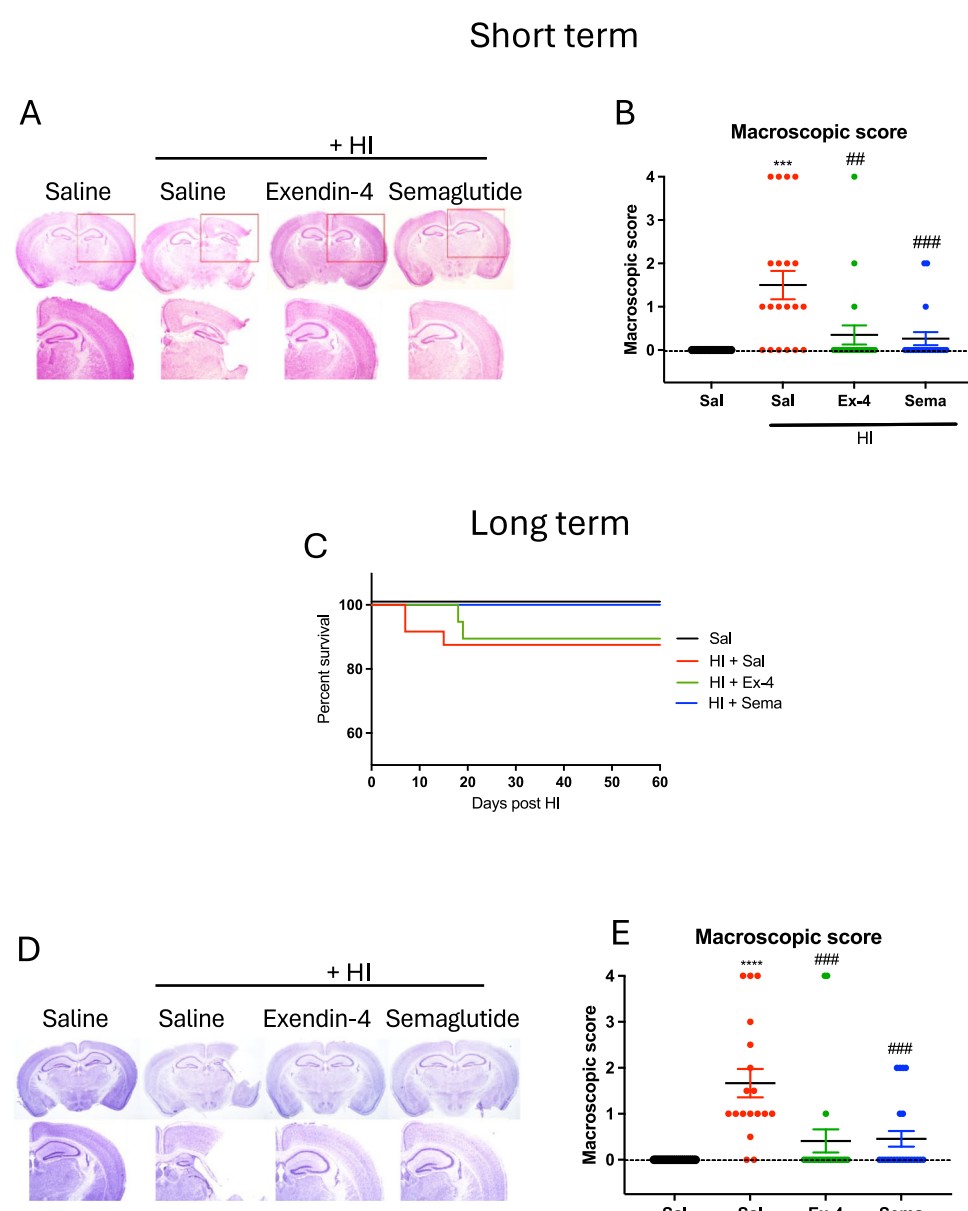

**Figure 4. Semaglutide provides single-dose, long-lasting protection after HI.**

(A) Nissl staining 7 days post-HI in control (saline, n = 16), HI alone (HI+saline, n = 18) and HI treated with exendin-4 (4× 0.5 μg/g every 12 h, n = 18) or semaglutide (1× 0.25 μg/g, n = 17) supplemented with glucose (magnification ×10, ×40). Evaluation of (B) macroscopic score 7 days post-HI. (C) Survival curve for saline (n = 18), HI saline-treated (n = 19), HI treated with exendin-4 (4× 0.5 μg/g) (n = 18) or semaglutide (1× 0.25 μg/g) supplemented with glucose (n = 18) followed for 8 weeks after HI injury. (D) Nissl staining (magnification ×10, ×40) of the cortex and hippocampus 8 weeks following HI, and (E) the macroscopic score of infarct volume for saline (n = 22), HI saline-treated (n = 18), HI treated with exendin-4 (4× 0.5 μg/g) (n = 22) or semaglutide (1× 0.25 μg/g) supplemented with glucose (n = 22). Data information: Error bars indicate mean ± SEM. Statistical analysis was performed using an ordinary one-way analysis of variance (ANOVA) corrected for using Tukey's multiple comparisons test. * or #, p < 0.05; ** or ##, p < 0.01; *** or ###, p < 0.001; **** or ####, p < 0.0001. * symbol indicates comparison with saline-treated controls, and # symbol indicates comparison with saline-treated HI group (HI + sal). Source data are available online for this figure.

potential long-term adverse haematological effects following treatment with exendin-4 or semaglutide, blood samples were taken after 2 months, and blood cell populations were measured. HI resulted in a significant increase in WBCs, platelets and haemoglobin distribution widths (HDWs) and a trend towards decreased RBCs, haemoglobin (HGB), increased red cell distribution widths (RDWs) and MCVs (Fig. EV5A). Mice treated with GLP1-R agonists improved the RBC,

RDW, HGB, HDW and MCV counts. Analysis of plasma parameters (sodium, chloride, urea, creatinine, total cholesterol, glucose, triglycerides, glycerol levels) was also performed but no significant difference was observed between groups except for a statistically significant increase in creatinine in all HI groups (Fig. EV5B). No microglia infiltration or activation was observed in organs harvested from experimental groups compared to controls (Fig. EV4A,B).

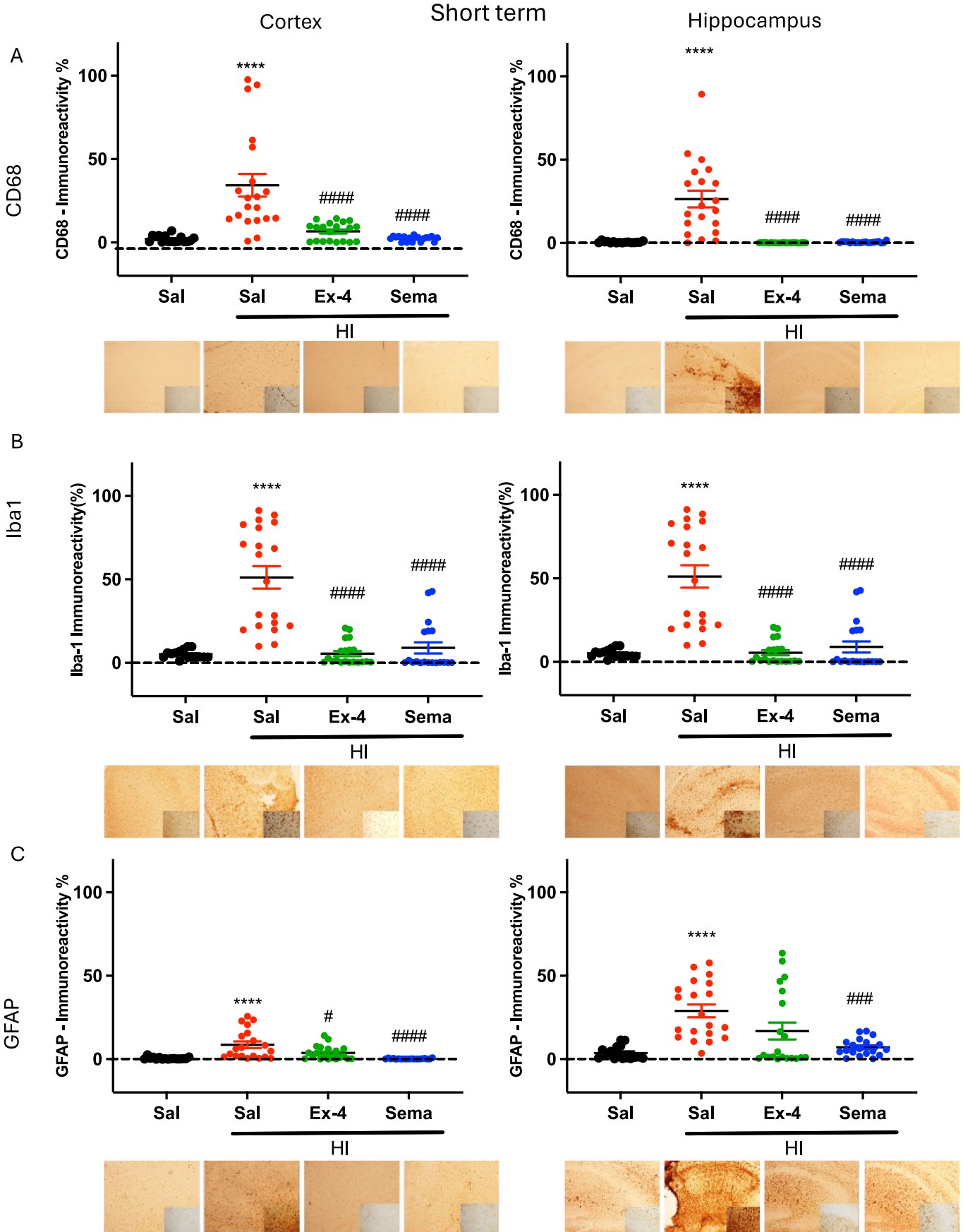

◄ **Figure 5. Semaglutide limits the neuroinflammatory response following HI in vivo 7 days post-HI.**

Quantitative immunoreactivity threshold measurements of markers for microglia (**A**) CD68, (**B**) Iba1, and (**C**) astrocytes (GFAP) in the cortex and the hippocampus 7 days post-HI in the 4 groups: saline, HI saline treated (HI+Sal), HI exendin-4 and HI semaglutide with cohort sizes of $n = 16$, $n = 18$, $n = 18$ and $n = 17$, respectively. Corresponding representative microscopic images taken at ×10 and ×40 magnification are included below the relevant graphs. Data information: Error bars indicate mean ± SEM. Statistical analysis was performed using an ordinary one-way analysis of variance (ANOVA) corrected for using Tukey's multiple comparisons test. * or #, $p < 0.05$; ** or ##, $p < 0.01$; *** or ###, $p < 0.001$; **** or ####, $p < 0.0001$. * symbol indicates comparison with saline-treated controls, and # symbol indicates comparison with saline-treated HI group (HI + sal). Source data are available online for this figure.

## Activation of GLP1-R modulates the CREB and PI3K/Akt pathway response following neonatal HI in vivo

Several studies have implicated the neuroprotective effects of GLP1-R agonists in modulation of the CREB and PI3/Akt kinase pathway. Previous studies have demonstrated that the Akt pathway is protective in neonatal HI (Brywe et al, 2005; Luo et al, 2019; Zhao et al, 2013). Therefore, we looked at activation of GLP1-R and the downstream effects on this pathway (Fig. 8A). HI had no significant effect on the levels of GLP1-R mRNA expression when measured by qPCR (Fig. 8B). However, the amount of GLP1-R mRNA was significantly increased in response to treatment with both exendin-4 (*$p = 0.0164$) and semaglutide (**$p = 0.0065$) in comparison to saline-administered mice. We investigated whether the ability of both GLP1-R agonists to modulate levels of CREB, as seen previously in vitro, was also occurring in vivo in P10 CD1 mice with induced HI injury. HI resulted in a downregulation of total CREB mRNA (**$p = 0.0075$) (Fig. 8C). Consistent with our observation in the in vitro neuronal cultures exposed to OGD, both GLP1-R agonists upregulated CREB mRNA in comparison to saline-treated HI mice ($p = 0.0008$ for exendin-4; $p = 0.0061$ for Semaglutide). All qPCR control Ct values are shown in Fig. EV6. Phosphorylated CREB (pCREB) is the transcriptionally active form of CREB (Fig. 8D,E) and phosphorylation of GSK3β on different sites modulates its enzymatic activity, with phosphorylation on Y216 being necessary for enzymatic activity, while phosphorylation of S9 amino acid renders GSK3β inactive. To investigate the active forms of CREB and GSK3β, we conducted western blots on brain lysates from each experimental group using antibodies specific to phosphorylation (Fig. 8D). The western blot showed significantly reduced levels of pCREB following HI (****$p < 0.0001$), which was increased following treatment with both GLP1-R agonists. Quantification of band intensities showed that this increase was statistically significant (###$p = 0.0001$ for HI Saline vs HI exendin-4 and ###$p = 0.007$ for HI Saline vs HI Semaglutide) (Fig. 8E). HI also increased pGSK3β-Y216 (****$p < 0.0001$) and decreased pGSK3β-S9 (****$p < 0.0001$) protein levels but were modulated following treatment with both GLP1-R agonists (Fig. 8D) and measured to be statistically significant following quantification relative to β-actin controls (###$p = 0.0001$ for HI saline vs HI exendin-4 and HI saline vs HI Semaglutide) (Fig. 8F). Analysis of overall PI3K and Akt protein levels in brain homogenates by western blot showed no differences between experimental groups (Fig. 8G). However, phosphorylated PI3K (pPI3K) was significantly reduced in the brains of mice that received HI compared with controls (p-PI3K, ****$p < 0.0001$). Following exendin-4 or semaglutide treatment, p-PI3K levels were almost restored to control levels (##$p = 0.0014$ and ##$p = 0.0010$) (Fig. 8G,H). Similarly, although total Akt remained unaffected, phosphorylated Akt (pAkt) was significantly reduced in mice that received HI compared with no HI control mice (*$p = 0.0190$) (Fig. 8G,I). However, phosphorylation levels were

maintained by exendin-4 (#$p = 0.0143$) or semaglutide treatment (##$p = 0.0070$) (Fig. 8G,I).

## Discussion

We have previously demonstrated that the GLP1-R peptide agonist exendin-4 is highly effective in protecting the brain in P7 CD1 mice with surgically induced HI (Rocha-Ferreira et al, 2018). The same study also showed that activating GLP1-R could be used in combination with hypothermia to further enhance therapeutic efficacy. However, the mechanisms behind how activation of GLP1-R provides protection to the brain in neonatal mice with HI brain damage is unknown. In this study, we aimed to further our understanding of cellular mechanisms triggered by GLP1-R activation that protect the brain and reduce or prevent infarct and inflammation. Furthermore, we introduced the clinically used T2DM treatment GLP1-R peptide agonist, semaglutide, to directly compare with exendin-4 to show that the effects of receptor activation on cellular mechanisms are conserved between different agonists. In addition, semaglutide has the advantage of having a significantly longer half-life (Knudsen and Lau, 2019).

The upregulation of the PI3/AKT signalling pathway following treatment with either agonist is likely a key component of the therapeutic efficacy. HI brain injury reduced mRNA and protein levels for phosphorylated PI3K and Akt, while treatment with both GLP1-R agonists normalised levels. This is supported by a number of other studies showing that GLP1-R agonists attenuate neuronal death via the GLP1-R/PI3K/Akt pathway in various models of neurodegeneration (Chen et al, 2016; Xie et al, 2018; Zhang et al, 2016; Zhu et al, 2016) and more recently in a study in neonatal rats with HI injury (Zeng et al, 2019). Current evidence suggests that activation of PI3K/Akt and its downstream pathways suppresses neuronal apoptosis in animals with HI brain injury (Tu et al, 2018) and OGD (Ye et al, 2019).

Activation of GLP1-R stimulates adenylyl cyclase signalling, leading to an increase in the levels of cAMP that activate cAMP response element-binding protein (CREB) (Zhang et al, 2016). We found that levels of mRNA and activated phosphorylated CREB were reduced in the brains of mice following HI injury. However, administration of both GLP1-R agonists led to upregulation in levels of CREB mRNA and protein phosphorylation. CREB is known to regulate the transcription of downstream Bcl-2 proteins (Meller et al, 2005). We showed that exendin-4 and semaglutide exhibited anti-apoptotic effects by modulating Bcl-2 family members. This is further supported by the significant reduction in macroscopic scores of the brain infarct. The reduction of ATF-3 in vivo, known to be upregulated following neuronal stress (Hunt et al, 2012), following GLP1-R agonist administration was

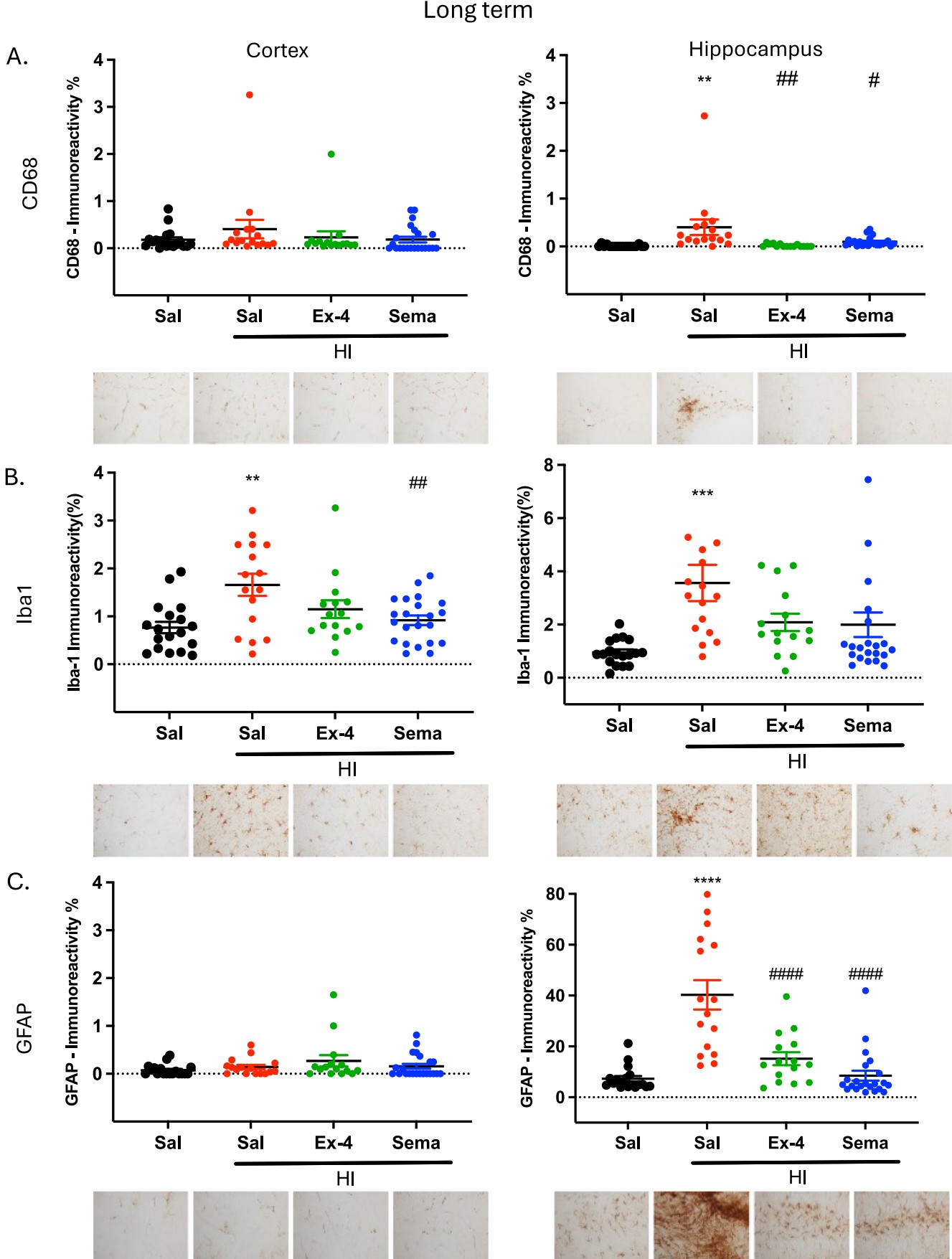

**Figure 6. Semaglutide limits the neuroinflammatory response following HI in vivo following long-term assessment.**

Quantitative immunoreactivity threshold measurements of markers for (A) macrophages (CD68), (B) microglia (Iba1), and (C) astrocytes (GFAP) in the cortex and the hippocampus 8 weeks post-HI in the 4 groups: saline, HI saline treated (HI+Sal), HI exendin-4 and HI semaglutide with cohort sizes of $n = 18$, $n = 19$, $n = 18$ and $n = 18$, respectively. Corresponding representative microscopic images taken at ×10 and ×40 magnification are included below the relevant graphs. Data information: Error bars indicate mean ± SEM. Statistical analysis was performed using an ordinary one-way analysis of variance (ANOVA) corrected for using Tukey's multiple comparisons test. * or #, $p < 0.05$; ** or ##, $p < 0.01$; *** or ###, $p < 0.001$; **** or ####, $p < 0.0001$. * symbol indicates comparison with saline-treated controls, and # symbol indicates comparison with saline-treated HI group (HI + sal). Source data are available online for this figure.

also observed in vitro in primary neuronal cultures exposed to OGD following treatment with exendin-4 and semaglutide. This would suggest that the health of these neurons is being preserved together with a reduction in HIF-1α.

An interesting observation was that a dose of 500 μg/kg of semaglutide in the P10 CD1 mice with surgically induced HI had no effect in reducing the infarct macroscopic score. Furthermore, some mice did not recover from HI surgery when administered with this dose. Conversely, the equivalent dose of exendin-4 was significantly therapeutic with no such issues observed. We reasoned that the high dose of semaglutide used in neonatal mice that had also received HI injury may be inducing hypoglycaemia, which exacerbates brain injury in the setting of neonatal HI in rodents (Vannucci and Yager, 1992). This was confirmed when blood glucose levels were found to drop significantly within 30 min of intraperitoneal administration. The reason for why semaglutide induces hypoglycaemia and the equivalent dose of exendin-4 is not fully understood and requires further investigation. However, hypoglycaemia was prevented through a combination of reducing the dose of semaglutide to 250 μg/kg and the addition of glucose supplementation, that successfully restored the therapeutic efficacy to that equivalent to exendin-4. The equivalent human dose would be approximately 1.22 mg in a 60 kg adult (0.02 mg/kg). This is approximately half of the FDA-recommended for obese adults (2.4 mg/kg) via weekly subcutaneous administration. Blood analysis revealed no abnormalities and no macrophage inflammation in major organs was detected. Furthermore, we confirmed that maximum activation of GLP1-R receptors in the brain was achieved more rapidly following an intraperitoneal administration of agonist compared to a subcutaneous administration in neonatal mice. Although the hypoglycaemia observed in these experiments was unexpected, it needs to be considered that we are using higher doses in neonatal mice that have received HI, via a route of administration that is not typically clinically used for the administration of semaglutide. Therefore, these factors taken together may cumulatively induce the observed hypoglycaemia.

Glycogen synthase kinase-3 β (GSK3β) is a multifunctional serine/threonine kinase, which is also regulated through PI3K/Akt activation inducing its inhibition (Duda et al, 2018). Inactivation of GSK3β by S(9) phosphorylation is implicated in mechanisms of neuronal survival and a neuroprotective role in various types of brain injury in animals (Farr et al, 2019; Valerio et al, 2011; Zeng et al, 2019; Zhao et al, 2017). Conversely, phosphorylation of a distinct site, Y(216), on GSK3β is necessary for its activity (Bhat et al, 2000) and leads to an increase in disease-induced neurodegeneration (Salcedo-Tello et al, 2011) and plays a strong proinflammatory role in many CNS diseases (Beurel et al, 2015). We observed that both GLP1-R agonists decreased the global GSK3β mRNA levels after HI and modulated phosphorylation of S9 and Y219 forms towards neuroprotection.

Neuronal damage is generally associated with neuroinflammation after brain injury (including ischaemic stroke (Jayaraj et al, 2019), traumatic brain injury (Wofford et al, 2019), intracerebral haemorrhage (Askenase and Sansing, 2016)) and neonatal HI injury is also associated with the activation of neuroinflammatory processes (Hagberg et al, 2015). Microglia play a critical role in neuroinflammation as the first line of defence whenever injury occurs. However, microglia can produce excessive proinflammatory mediators that exacerbate brain damage and affect the levels of anti-inflammatory mediators such as IL-10, which is expressed as part of the brain repair response (Garcia et al, 2017). Our data show that microglial- and astroglial-mediated inflammatory responses are triggered by HI injury to the brain. However, treatment with both GLP1-R agonists significantly ameliorated the microglial- and astroglial-mediated inflammatory response. It remains unclear whether the reduced inflammation is a direct consequence of the drugs acting primarily on glial cells, or whether it is a consequence of neuroprotection, or both. We have previously shown that GLP1-R is present on neurons and astrocytes at this early stage of development (Rocha-Ferreira et al, 2018). Although the reduction in infarct size and programmed cell death may suggest that the amelioration in inflammatory response is a secondary response, the precise sequence of events requires further investigation.

We have previously reported on the benefits of exendin-4 treatment in P7 CD1 mice with surgically induced HI 48 h after injury. Here, we assessed mice up to 8 weeks of age that received the 4 doses of exendin-4 in the first 48 h and a single dose of semaglutide after HI injury. This also allowed us to assess survival and locomotor function. At 8 weeks of age, macroscopic scoring of infarct volumes remained significantly reduced in treatment groups when compared to the untreated HI group and neuroinflammation was improved. Both GLP1-R agonists also improved survival and locomotor function, that is known to be impaired in neonatal rats with HI (Borjini et al, 2019). Neuroinflammation remains significantly reduced at 8 weeks.

An important consideration is that in this study the GLP1-R agonists were administered immediately following HI injury. In clinical practice, this is not practical; for example in the large trials of therapeutic hypothermia, the average delay was over hours and many took significantly longer (Mathew et al, 2022). Thus, the present study establishes proof of principle but further studies of the window of opportunity are now essential. However, we have previously demonstrated that the therapeutic window of opportunity using a GLP1-R agonist (exendin-4) can still be significantly effective in ameliorating HI injury in the brain even when administered at 4 h post injury (Rocha-Ferreira et al, 2018). A major limitation of the present study is that we did not measure the pups' temperatures after returning them to the dam. Previous studies confirm that after HI, rodent pups consistently become hypothermic (Wood et al, 2018) and so we cannot rule out the

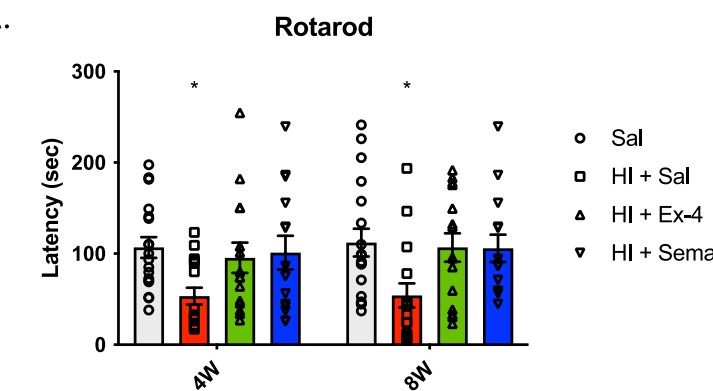

A. **Rotarod**

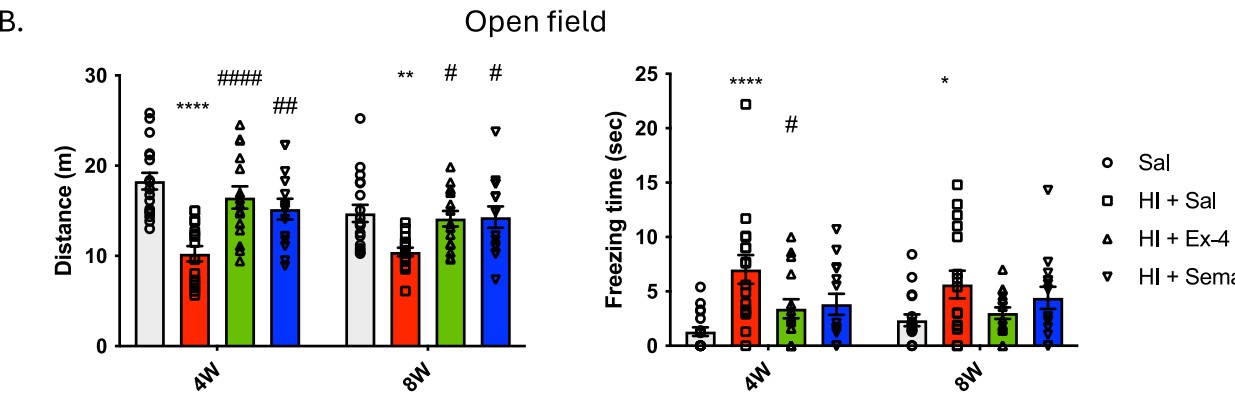

B. Open field

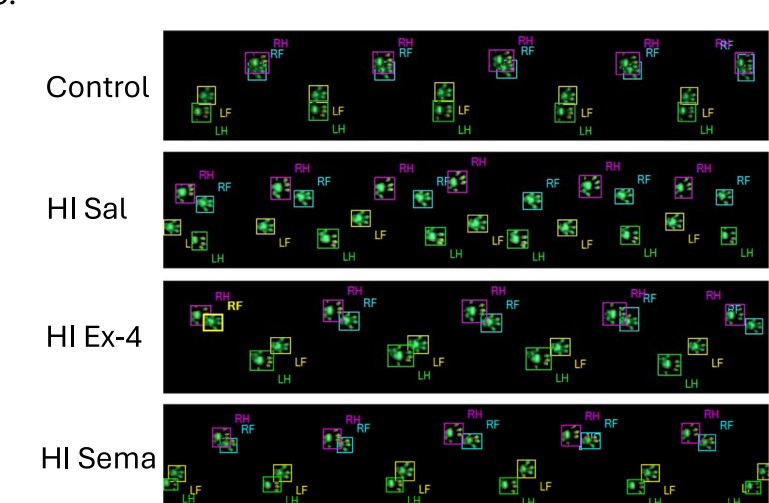

C.

Control

HI Sal

HI Ex-4

HI Sema

possibility that the apparent treatment effects were exaggerated by drug induced hypothermia (Klahr et al, 2017).

## Conclusion

Taken together, our data demonstrate that GLP1-R agonists can trigger a number of mechanisms to achieve neuroprotection and reduce the inflammatory response in vitro and in vivo following HI. This is linked to the enhanced activity of the PI3K/AKT pathway and increased levels of cAMP in response to GLP1-R activation. There is significant scope for more in-depth studies of any of the mechanistic outcomes that we have reported on. Furthermore, we show that another GLP1-R agonist, semaglutide, is able to ameliorate HI damage from a single injection in a similar way to exendin-4 which requires 4 doses over the

**Figure 7. GLP1-R agonists improve behavioural outcomes following HI in vivo.**

Locomotor functions evaluated at 4 and 8 weeks of age with (A) rotarod latency to fall in control (saline, $n = 18$), HI alone (HI+saline, $n = 16$) and HI animals treated with exendin-4 (4× 0.5 μg/g every 12 h, $n = 15$) or semaglutide (1× 0.25 μg/g, $n = 14$) supplemented with glucose. (B) open field distance travelled in control (saline, $n = 18$), HI alone (HI+saline, $n = 16$) and HI animals treated with exendin-4 (4× 0.5 μg/g every 12 h, $n = 15$) or semaglutide (1× 0.25 μg/g, $n = 14$) supplemented with glucose and (C) freezing time in control (saline, $n = 18$), HI alone (HI+saline, $n = 16$) and HI animals treated with exendin-4 (4× 0.5 μg/g every 12 h, $n = 15$) or semaglutide (1× 0.25 μg/g, $n = 14$) supplemented with glucose. (C) Representative images of paw prints captured using the Noldus CatWalk XT automated gait analyser. The individual paws are recognised using distinctive colours. Data information: Error bars indicate mean ± SEM. Statistical analysis was performed using an ordinary two-way analysis of variance (ANOVA) corrected for using Tukey's multiple comparisons test. * or #, $p < 0.05$; ** or ##, $p < 0.01$; *** or ###, $p < 0.001$; **** or ####, $p < 0.0001$. * symbol indicates comparison with saline-treated controls, and # symbol indicates comparison with saline-treated HI group (HI + sal). Source data are available online for this figure.

important initial 48 h time period after HI injury. Hypoglycaemia was observed in neonatal mice administered with high doses of semaglutide via intraperitoneal injection, but this could be controlled by reducing the dose and supplementation with glucose. This study provides further evidence for the potential of GLP1-R agonists to be a clinical option for the treatment of HIE in their own right, or possibly in combination with hypothermia.

# Methods

## Study design

The objective of the study was to understand the mechanism and pathways induced by treatment with GLP1-R agonists and to assess the efficacy of these drugs in hypoxic-ischaemic encephalopathy, both in vitro using cells that have undergone OGD and in vivo in P10 CD1 mice that have had induced HI, with short-term observations (7 days post-insult) and long-term observations performed (2 months post-insult – 2 M). The control and treatment groups and the number of biological replicates (sample sizes) for each experiment are specified in the figure legends. For the in vivo studies, animals were randomly allocated by one researcher to the control and treatment groups, with identifying marks for each experimental cohort. The next researcher was blinded to the identity of each group for behavioural analysis. The animals were housed together to minimize environmental differences and experimental bias. Sample size was based on 5% significance with 80% power with one mouse representing an experimental unit. Effect size was estimated from our recent use of the animal model by Rocha-Ferreira et al (Rocha-Ferreira et al, 2018) comparing HI (Veh) with HI (Treatment) groups and calculations performed using publicly available PS: Power and Sample Size Calculation v3.1.6 software. No animals were excluded during the experiment.

## CHO cell culture and luciferase assay

For biological evaluation of GLP-1/Fc for human GLP1-R activation, GLP1-R/pCRE-Luciferase-co-expressed CHO cell lines were established using the lipofectamine method. Human GLP1-R/pCRE-luciferase-co-expressed CHO cell line express both human GLP1-receptor and CRE-luciferase (Kim et al, 2009). Using a luciferase assay system (Promega – E1500), activity of the GLP1 receptor can be assessed following treatment with GLP1-R agonists for 24 h in 96-well plates.

## Western blot

Tissues were homogenized (Ultra-Turrax TP, IKA Labortechnik, Wasserburg, Germany) on ice in 300 ml of RIPA lysis buffer (Thermo) per 100 mg of tissue with 1X protease inhibitor cocktail (Thermo) and incubated for 30 min. Lysates were centrifuged at $14{,}000 \times g$, 4 °C for 20 min and overall protein concentrations of the supernatant were determined by Pierce BCA Protein Assay (Life Technologies). Samples were incubated at 37 °C for 30 min in 1X LDS sample buffer (Life Technologies) and 1X sample reducing agent (Life Technologies), after which 40 μg of protein were loaded per well in a NuPAGE Bis–Tris 4–12% polyacrylamide gel for protein separation via SDS-PAGE electrophoresis. Proteins were transferred to PDVF membrane at 400 mA for 1 h and membrane was blocked for 1 h at 4 °C with 5% BSA in TBS with 3% Tween-20. Membranes were subsequently incubated overnight at 4 °C with primary antibodies (Appendix Table S2) with 3% BSA in TBS with 3% Tween-20. After 3 washes in TBS, antibody staining was revealed using HRP-conjugated goat anti-rabbit IgG incubated for 2 h at room temperature in TBS with 3% Tween-20 with 3% BSA. Blots were developed with the ECL system (SuperSignal West Pico, Life Technologies) and imaged using a Genegnome imager (Syngene, Cambridge, UK).

## Primary neuronal cultures exposed to oxygen–glucose deprivation (OGD)

Animal use was in accordance with local rules (UCL, London) and with the regulations and guidance issued under the Animals (Scientific Procedures) Act (1986). Pregnant CD1 mice were sacrificed by schedule 1 methods at 13.5–14.5 days of gestation and embryonic cortical neurons prepared as described previously (Thornton et al, 2011). Cells were plated at a density of $2 \times 10^6$ cells/6 cm plate and propagated. At DIV 10–12, neurons were subjected to oxygen–glucose deprivation (OGD) in neurobasal-A medium lacking glucose and incubated in a hypoxia chamber (Billups-Rothenburg Inc., Del Mar, CA, USA) filled with an anoxic atmosphere of 5% $CO_2$ balanced in nitrogen at 37 °C for 2 h. Pharmacological treatments (exendin-9, exendin-4, Semaglutide) were incubated with the cells after OGD (OGD experiments) for 2 h. Exendin-9 is an antagonist of the GLP1-R receptor and used in these to study to block the receptor and confirm inhibition of the exendin-4 and semaglutide binding. Culture medium was assayed after 24 h.

### MTT—cell viability test
Culture medium was assayed after treatment at 24 h post-OGD. All samples were processed using MTT assay kit as per manufacturer's instructions (Abcam – Ab211091).

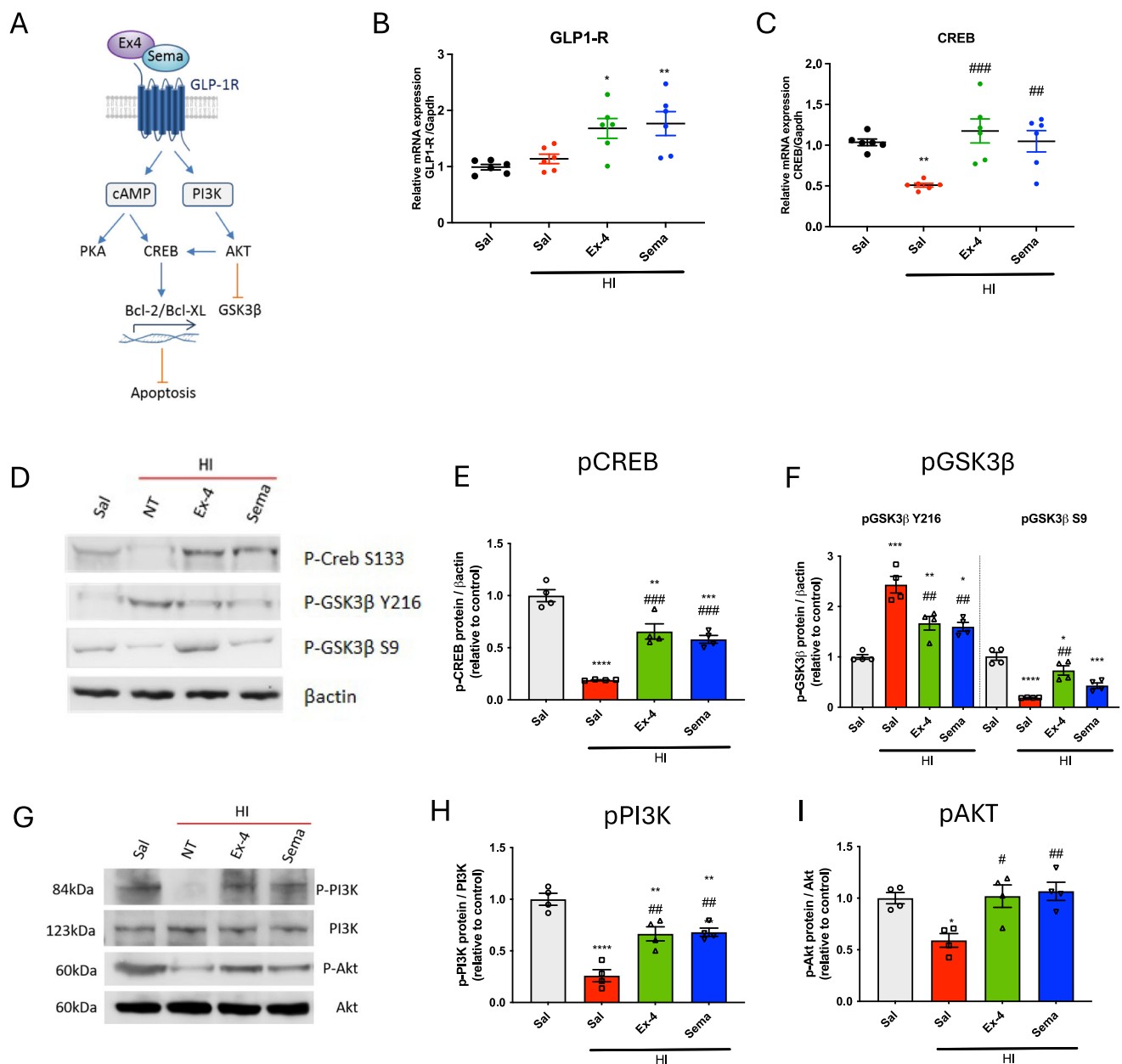

**Figure 8. GLP1-R agonists activate CREB and PI3K pathways following neonatal HI in vivo.**

(A) GLP1-R agonists Exendin-4 and Semaglutide trigger GLP1-R signalling via cAMP and PI3K. Transcription of prosurvival genes and inhibition of apoptotic pathways may counteract the effect of hypoxia-mediated apoptosis. Incubation with GLP1-R antagonist Exendin 9 prevents GLP1-R activation. (B) GLP1-R gene expression levels assessed with qPCR ($n = 6$ for all experimental groups). (C) Evaluation of neuroprotection mechanisms with measurement of gene expression for CREB ($n = 6$ for all experimental groups) and (D, E) protein levels for phosphorylated-CREB ($n = 4$ for all experimental groups) and (D, F) phosphorylated-GSK3β ($n = 4$ for all experimental groups). Activation of the PI3/Akt pathway was also assessed by western blot (G) quantification of phospho-PI3K (H) and (G, I) phosphorylated-Akt protein levels ($n = 4$ for all experimental groups). Data information: Error bars indicate mean ± SEM. Statistical analysis was performed using an ordinary one-way analysis of variance (ANOVA) corrected for using Tukey's multiple comparisons test. * or #, $p < 0.05$; ** or ##, $p < 0.01$; *** or ###, $p < 0.001$; **** or ####, $p < 0.0001$. * symbol indicates comparison with saline-treated controls, and # symbol indicates comparison with saline-treated HI group (HI + sal). Source data are available online for this figure.

## Caspase assays

Culture medium was assayed after treatment at 24 h post-OGD. All samples were processed using Multiplex caspases 3, 8 and 9 activity assay kit as per manufacturer's instructions (Abcam – Ab219915).

## Animals

All animal experiments and care protocols were carried out according to local guidelines through the UCL Animal Welfare and Ethical Review Board, the UK Animals (Scientific Procedures) Act 1986 and approved by the Home Office (PCC436823). The

animals were group-housed housed in individually ventilated cages (IVCs) including environmental enrichment, in a temperature- and light-controlled environment, with food and water given ad libitum. Light, temperature and cage location were constantly maintained to reduce confounding variables. Predetermined parameters were established as the human endpoints. These included: a loss of 10% body weight, changes in appetite, unrelieved pain/distress, organ system dysfunction or failure and clinical signs which indicate that the animal has entered a moribund state. The ARRIVE guidelines for reporting were followed.

## Surgically induced hypoxia-ischaemia in P10 CD1 mice and treatments

All experiments involved litters of postnatal day 10 CD1 mice (P10) bred in-house. The CD1 strain was used due to their large litter sizes and excellent maternal instincts. This is advantageous where pups are separated from their mothers for extended periods of time. The surgical procedures were performed as previously described (Kendall et al, 2012; Lange et al, 2014; Rocha-Ferreira et al, 2018). Body temperature was maintained during surgery and incubations to prevent hypothermia that can confound therapeutic readouts and the effect of drugs (Klahr et al, 2017). This was achieved using heating mats, heated incubators and prompt returning of pups to the dam. Briefly, male and female P10 mice were anesthetized with isoflurane (5% induction and 1.5% maintenance). The left common carotid artery was permanently occluded with 8/0 polypropylene suture and the wound closed with tissue glue. The body temperature was maintained using a heating mat. The mice recovered in an incubator at 37 °C and were returned to the dam for 1 h. The pups were then placed in a hypoxia chamber and exposed to humidified 8% oxygen/92% nitrogen (3 L/min) at 37 °C for 20 min, resulting in moderate to severe brain damage (Lange et al, 2014; Rocha-Ferreira and Hristova, 2016). The pups were then immediately returned to the dam. The P10 rodent induction of HI through unilateral occlusion of the carotid presents phenotypical similarities to the grey and white matter injury observed in humans, i.e., tissue loss, cell-death, microglial-mediated immune response, and astrogliosis as well as alteration in neurobehavioral performance (Vannucci and Vannucci, 2005). 1% mortality was observed following this procedure. All treatments were administered immediately after HI via intraperitoneal administration. Four doses of exendin-4 were administered in total, one every 12 h (each dose of 0.5 μg/g). Animals were randomized for short-term (ST) and long-term (LT) studies to: (i) saline ($n = 16$ ST; $n = 18$ LT); (ii) hypoxic-ischaemic group with saline treatment ($n = 18$ ST; $n = 19$ LT); (iii) hypoxic-ischaemic group with four high doses of exendin-4 (0.5 μg/g) (ip) administered every 12 h, starting immediately after HI ($n = 18$ ST; $n = 18$ LT); (iv) hypoxic-ischaemic group with a single dose of semaglutide (0.25 μg/g) (ip) administered immediately after HI ($n = 17$ ST; $n = 18$ LT). Treated animal were culled at either 48 h post HI for ST studies or 60 days post HI for LT studies. The study design is illustrated in Fig. EV1.

## Collection of tissues and infarct volume measurement

Experimental cohorts were culled at either short-term or long-term time points. The mice were perfused with 4% paraformaldehyde and then transferred to 30% sucrose before being snap-frozen. 40

μm coronal brain sections were collected starting from where the corpus callosum fused and kept in TBSAF solution. Macroscopic score of tissue loss was also conducted using the following scale during the tissue collection; 0 presenting no visible damage; 1—25%, 2—50%, 3—75% and 4—100% hemispheric loss or found dead during the study.

## Quantitative polymerase chain reaction

RNA from neuron cultures or brains was extracted and isolated using RNeasy Mini kit (Qiagen) and first-strand cDNA was generated using High-Capacity cDNA Reverse Transcription kit (Applied Biosystems). Quantitative RT-PCR was carried out using a StepOnePlus (Applied Biosystems) with the SsoAdvanced™ Universal SYBR Green PCR Core Reagents Supermix (Bio-Rad). Primers sequences were designed to detect GLP1-R, CREB, GSK3β, il-10, IL-4, IL-1β, ATF-3, Bcl-2, Bcl-xL (Appendix Table S1). Data from StepOne™ software v2.3 were calibrated to Gapdh, and the relative quantitation of gene expression was performed using the comparative CT method.

## Staining analysis

At the end of the study (P10 + 7 days for short-term analysis and P10 + 2 months for long-term analysis), mice were sacrificed. The brains and other organs (heart, lung, liver, pancreas, spleen and kidney) were fixed via cardiac perfusion with phosphate-buffered saline (PBS), dissected out and postfixed at 4 °C in 4% paraformaldehyde (PFA) for at least 24 h and cryoprotected in 30% sucrose solution overnight. All staining except H&E were performed on 40-μm-thick free-floating brains and organs sections.

## Immunofluorescence

Neuronal cultures ($n = 6$ per group) were stained by simultaneous overnight incubation with ATF-3 and anti-NeuN antibodies (Appendix Table S2). Representative images were captured using a Zeiss LSM710 confocal microscope and Zen software (Carl Zeiss AG). Sections were treated with fluorochrome-conjugated secondary antibody (1:1000, Alexa Fluor 488 and 546) and mounted with a Fluoromount-GTM mounting medium (ThermoFisher Scientific). Representative images were captured using a Leica SP5 confocal microscope equipped with LAS-AF software. All images were processed using Imaris v.9.1 (Bitplane AG).

## Immunohistochemistry

After pre-treatment in 1% $H_2O_2$ and 15% normal serum (Sigma-Aldrich), sections were stained overnight using anti-CD68/NeuN/GFAP/Iba-1 antibodies (Appendix Table S2) with 10% normal serum in TBS-T. Sections were then incubated with biotinylated secondary antibodies (dilution 1:1000, Vector Laboratories) and subsequently processed using avidin-biotinylated peroxidase (Vectastatin ABC kit, dilution 1:1000, Vector Laboratories). Sections were subsequently mounted, coverslipped and imaged with a laser scanning confocal microscope (Zeiss LSM 710, Carl Zeiss AG, Cambridge, UK). Quantification was performed with Image Pro Premier software.

## H&E staining

The brain and organ sections were stained with H&E for the histological studies. Tissue sections were mounted on chrome-gelatine-coated slides and left to air-dry overnight. The sections were then stained, while protected from light, with 0.1% Mayer Haematoxylin (Sigma-Aldrich, Missouri, USA) for 10 min. The slides were rinsed in distilled water before incubation in 0.5% eosin solution (Sigma-Aldrich, Missouri, USA). The sections were washed and then dehydrated for 30 s each in increasing concentrations of ethanol (50%, 75%, 95%, 100%). Finally, the slides were incubated in Histoclear for 30 min and coverslipped with DPX mountant medium.

## Rotarod test

Motor coordination and balance were assessed using an accelerating rotarod (Panlab, Harvard Apparatus, Cambridge, UK). Mice were first habituated to the rotating rod at a constant speed of 4 rpm for 300 s (5 min), during which time latency to fall was not recorded. The habituation trial was only performed on the first day. Mice were subsequently exposed to a rotating rod starting at 4 rpm and linearly accelerated to 40 rpm over a 5 min period. Three trials were realized per day with a maximum time of 5 min.

## Open field test

The Open Field activity monitoring system comprehensively assesses locomotor and behavioural activity levels in mice. The apparatus is constructed of clear Plexiglas and measured $72 \times 72$ cm with 36 cm walls. Each mouse is placed in a corner of the open field and allowed to explore the apparatus freely for 5 min. Measures of total distance travelled, duration of immobility and mobility, average speed were obtained with an automated camera-based computer tracking software (AnyMaze, Rathmines, Ireland). Following the 5 min test, mice were returned in their home cages and the open field was cleaned with 70% of Ethanol and permitted to dry between tests. To assess the process of habituation to the novelty of the arena, mice were exposed to the apparatus for 5 min on 2 consecutive days.

## Catwalk XT gait analysis

Automated gait analysis was performed using the CatWalk XT system (Noldus, Wageningen, The Netherlands), where mice were filmed walking a minimum of five times across a backlit stage at 1- and 2-months post-insult. Runs were assigned and analysed using the CatWalk XT software v9.1 (Noldus) to produce footprint, stride, and overall run measurements. Parameters measured include stride length (the distance between successive paw placement of the same paw), average speed, swing speed (speed of the paw between successive paw placement), regularity index (% index for the degree of interlimb coordination during gait), step sequence (% of steps following normal alternative step pattern), body speed (distance covered per second), single support (relative contact duration of a single paw with the glass surface), lateral support (relative contact duration of lateral paws with the glass surface) and girdle support (relative contact duration of girdle paws with the glass surface).

## Blood and plasma analysis

Postnatal Day 10 (P10) mice underwent one injection of semaglutide (0.25 µg/g per dose, Bachem) via intraperitoneal administration. Toxicity study compared saline- ($n = 6$ per group) and semaglutide-injected P10 mice. Seven days after the last injection, blood samples were taken via cardiac puncture and collected in EDTA-coated tube. The analysis was performed by MRC Harwell Clinical Pathology laboratory (Mary Lyon Centre, UK). Blood and plasma analysis were performed 2 months post-insult for the long-term study. Various parameters were measured from blood samples, including: total white blood cells, neutrophils, lymphocytes, monocytes, eosinophils and basophils counts, hematocrit (HCT), platelets, red blood cells (RBCs), haemoglobin and mean corpuscular volume (MCV). Plasma samples were assayed for various parameters, including: sodium, chloride, urea, creatinine, inorganic phosphate, alkaline phosphatase (ALP), alanine aminotransferase (ALT), total protein, albumin, total cholesterol, high-density lipoprotein (HDL), low-density lipoprotein (LDL), glucose, triglycerides, glycerol, free fatty acids (FFA), amylase, creatine kinase (CK) and fructosamine (FRUCT). Additional parameters such as potassium, aspartate aminotransferase (AST), iron, bilirubin & lactate dehydrogenase (LDH) were not reported after analysis, due to the interference of haemolysis with various assays, affecting reliability.

## Blood glucose analysis and cyclic AMP analysis

Blood glucose levels (mmol/l) were measured using a blood glucose monitor (CodeFree, SD Biosensor) in naive controls and mice following one semaglutide dose administration (0.25 µg/g) with or without glucose supplementation, intraperitoneally or subcutaneously to determine the best route of administration. Blood samples were collected via cardiac puncture at 0.5 h, 2 h, 24 h, 7 days, and 14 days for glucose test. Brains of P10 naive mice or treated at the different time points were collected ($n = 4$ per group). All samples were processed using AMP direct immunoassay kit as per manufacturer's instructions (Abcam).

## Statistical analysis

Analyses were performed using GraphPad Prism 7.0 software. No animals were excluded from the analysis. Data from two groups or more than two independent variables were analysed by ANOVA, followed by the relevant post hoc test. For comparison between two groups, an unpaired $t$-test was performed. Normality and equal variance were assessed and non-parametric analysis performed if relevant. Data are presented as means ± SEM. Significance levels between controls and treated groups were set when $p < 0.05$ (* or #), $p < 0.001$ (** or ##), $p < 0.001$ (*** or ###) and $p > 0.0001$ (**** or ####) between different conditions: * used to compare with control group (no treatment) and # used to show the impact of the treatment.

## For more information

Action Medical Research updates on this project: https://action.org.uk/research/brain-damage-birth-could-diabetes-medicine-be-protective.

## The paper explained

### Problem

Hypoxia-ischaemic encephalopathy is a devastating condition involving injury to the brain due to a lack of blood flow and reduced oxygen. It commonly effects newborns due to complications in childbirth or with the umbilical cord. Depending on the severity of the brain damage, symptoms range from life-long disabilities to death. This study investigated how drugs commonly used to treat diabetes can prevent the brain damage in a mouse model of hypoxia-ischaemic encephalopathy.

### Results

A single injection of a diabetes drug effectively reduced brain damage and significantly improved disabilities and survival of the treated mice. The drugs activated pathways in cells of the brain that protect them from irreversible damage and death. This also reduced inflammation and cumulatively improved outcomes in the mouse model of hypoxic-ischaemic encephalopathy.

### Impact

The study showed that drugs that are already safely in use for treating diabetics could potentially be repurposed as a treatment for hypoxic-ischaemic encephalopathy. Furthermore, through our increased understanding of how these drugs do this at a cellular level, there is also potential for their use in other severe conditions that affect the brain and for which there are no effective treatments available.

## Data availability

This study includes no data deposited in external repositories.

The source data of this paper are collected in the following database record: biostudies:S-SCDT-10_1038-S44321-024-00079-1.

## Peer review information

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

## Acknowledgements

This work was funded by Action Medical Research (GN2485). AAR is also supported by the UK Medical Research Council (MR/N026101/1, MR/R025134/1, MR/S009434/1, MR/S036784/1, and MR/T044853/1), Wellcome Trust Institutional Strategic Support Fund/UCL Therapeutic Acceleration Support (TAS) Fund (204841/Z/16/Z), European Union's Horizon 2020 research and innovation programme under grant agreement no. 666918 and the NIHR GOSH BRC (the views expressed are those of the author(s) and not necessarily those of the NHS, the NIHR, or the Department of Health). We thank Drs Giovanni DiPasquale and John Chiorini (National Institutes of Health, USA) for providing the GLP1R/pCRE-luciferase/CHO-K1 cells.

## Author contributions

**Laura Poupon-Bejuit:** Data curation; Formal analysis; Investigation; Methodology; Writing—original draft. **Amy Geard:** Data curation; Investigation; Methodology; Writing—original draft; Writing—review and editing. **Nathan Millicheap:** Data curation; Investigation. **Eridan Rocha-Ferreira:** Writing—original draft; Writing—review and editing. **Henrik Hagberg:** Funding acquisition; Writing—original draft; Writing—review and editing. **Claire Thornton:** Conceptualization; Formal analysis; Funding acquisition; Writing—original draft; Writing—review and editing. **Ahad A Rahim:** Conceptualization; Formal analysis; Supervision; Funding acquisition; Investigation; Methodology; Writing—original draft; Project administration; Writing—review and editing.

Source data underlying figure panels in this paper may have individual authorship assigned. Where available, figure panel/source data authorship is listed in the following database record: biostudies:S-SCDT-10_1038-S44321-024-00079-1.

## Disclosure and competing interests statement

The authors declare no competing interests.

# Expanded View Figures

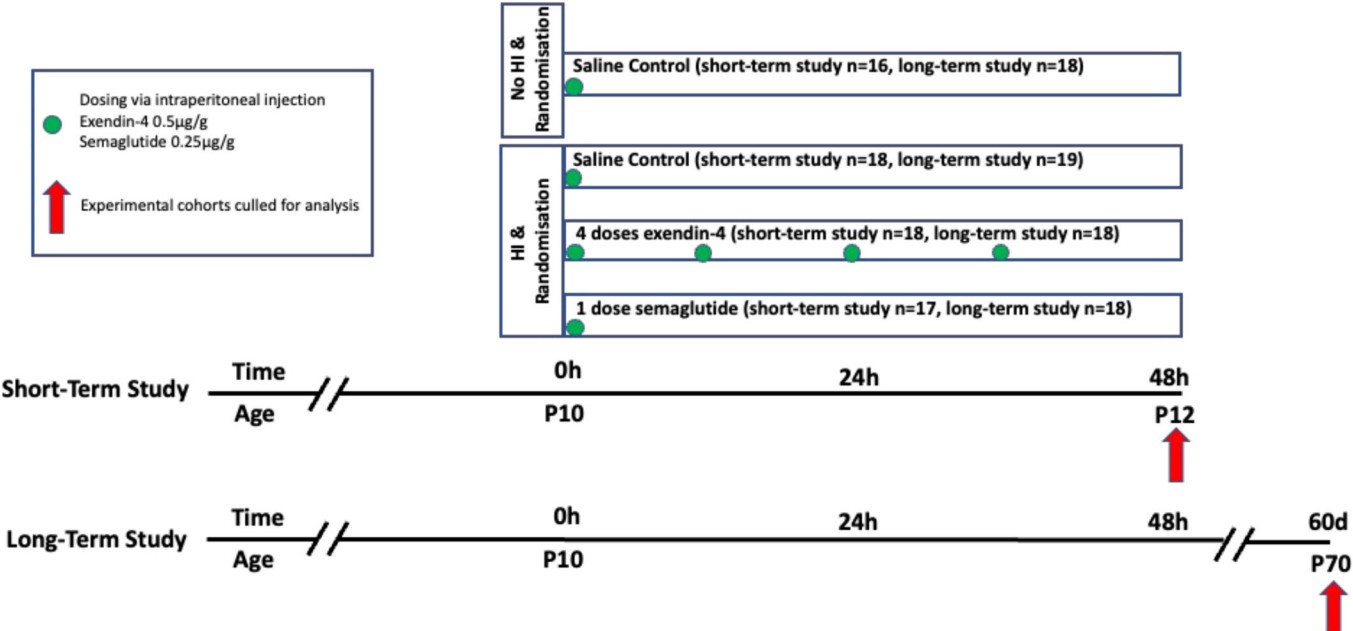

**Figure EV1.**  In vivo HI and GLP1-R agonist short-term and long-term studies experimental design.

## In vitro data

### Apoptosis

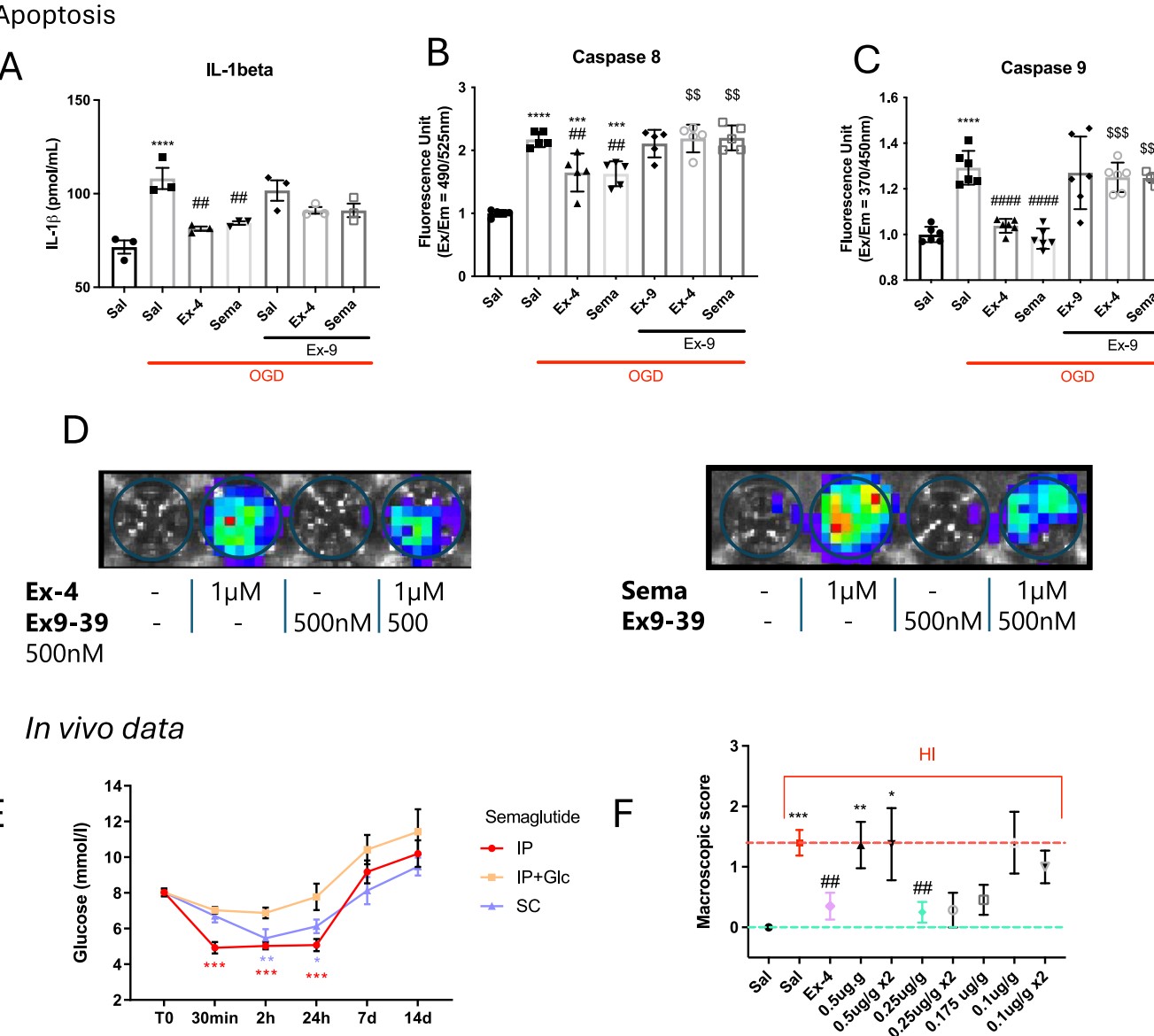

## In vivo data

**Figure EV2. Complementary results for in vitro study post-OGD and in vivo study post-HIE.**

(A) Evaluation of proinflammatory marker IL-1β using qPCR in the in vitro neuronal cells exposed to OGD following treatment with exendin-4 or semaglutide, and in combination with exendin-9 (Ex-9) ($n = 3$ for all experimental groups). (B) Apoptotic proteins caspase 8 ($n = 5$ for all experimental groups) and (C) caspase 9 ($n = 6$ for all experimental groups) were measured using caspase assays. (D) Luminescent images of GLP1R/pCRE-luciferase/CHO-K1 cells treated with 0–1 μM of exendin-4 or semaglutide with or without antagonist, exendin9-39 (500 nM). (E) Determination of optimal routes of administration for avoiding hypoglycaemia with glucose measurement after semaglutide IP, semaglutide SC and semaglutide + glucose regimens ($n = 4$ per experimental group for each time point). (F) Dosing experiment with evaluation of macroscopic score after HI and treatment to determine optimal dose for semaglutide in the 10 groups: Sal ($n = 20$), HI + Sal ($n = 44$), HI + Ex-4 ($n = 20$), HI + 0.5 μg/g semaglutide ($n = 18$), HI + 2× 0.5 μg/g semaglutide ($n = 8$), HI + 0.25 μg/g semaglutide ($n = 16$), HI + 2× 0.25 μg/g semaglutide ($n = 7$), HI + 0.175 μg/g semaglutide ($n = 11$), HI + 0.1 μg/g semaglutide ($n = 5$), HI + 2× 0.1 μg/g semaglutide ($n = 11$). Data information: Error bars indicate mean ± SEM. Statistical analysis was performed using an ordinary one-way (A–C) or two-way (E, F) analysis of variance (ANOVA) corrected for using Tukey's multiple comparisons test. * or # or $, $p < 0.05$; ** or ## or $$, $p < 0.01$; *** or ### or $$$, $p < 0.001$; **** or #### or $$$$, $p < 0.0001$. * to compare with control (Sal) and # to compare with OGD group (Sal + OGD). $ symbol used to compare GLP1-R agonists treatment with corresponding treatment in combination with exendin-9.

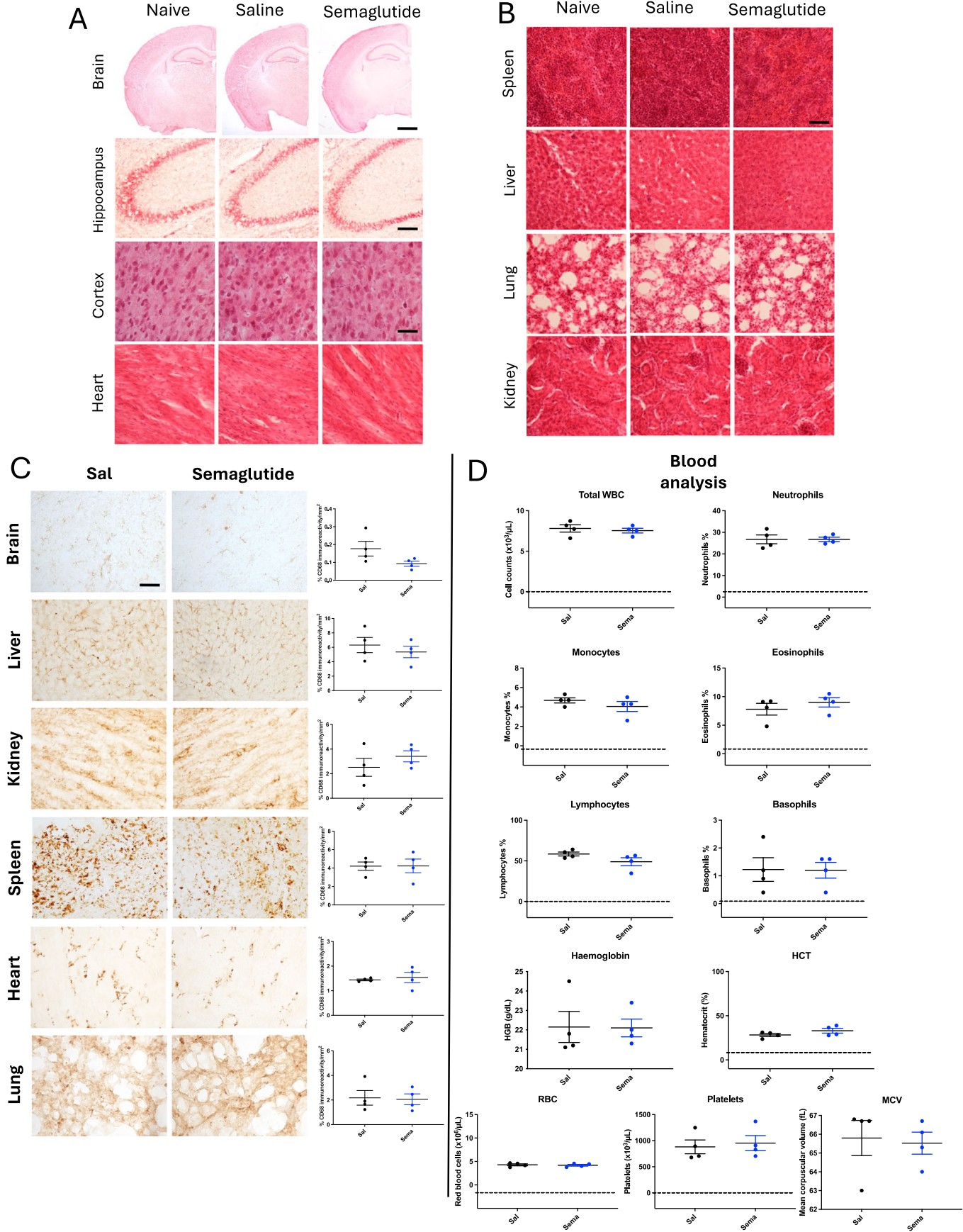

**Figure EV3.   Toxicity assessment after semaglutide treatment.**

(**A, B**) Representative images of Haematoxylin & Eosin staining showing no adverse effects observed in tissue architecture in naïve P10 mice following a single high dose of semaglutide (0.25 µg/g) in the brain or major visceral organs compared to controls. (**C**) CD68 staining and quantification revealed no macrophage activation or infiltration compared to controls. (**D**) Blood analysis also showed no significant changes in response to high dose semaglutide ($n = 4$ for all experimental groups). Scale bar: 0.1 cm (brain), 100 µm (Hippocampus), 60 µm (Cortex, Heart, Spleen, Liver, Lung, Kidney and all CD68 staining images in panel **C**). Data information: error bars indicate mean ± SD, Statistical analysis was performed using a t-test. *$p < 0.05$, **$p < 0.01$, ***$p < 0.001$, ****$p < 0.0001$. * to compare with control and group treated with semaglutide.

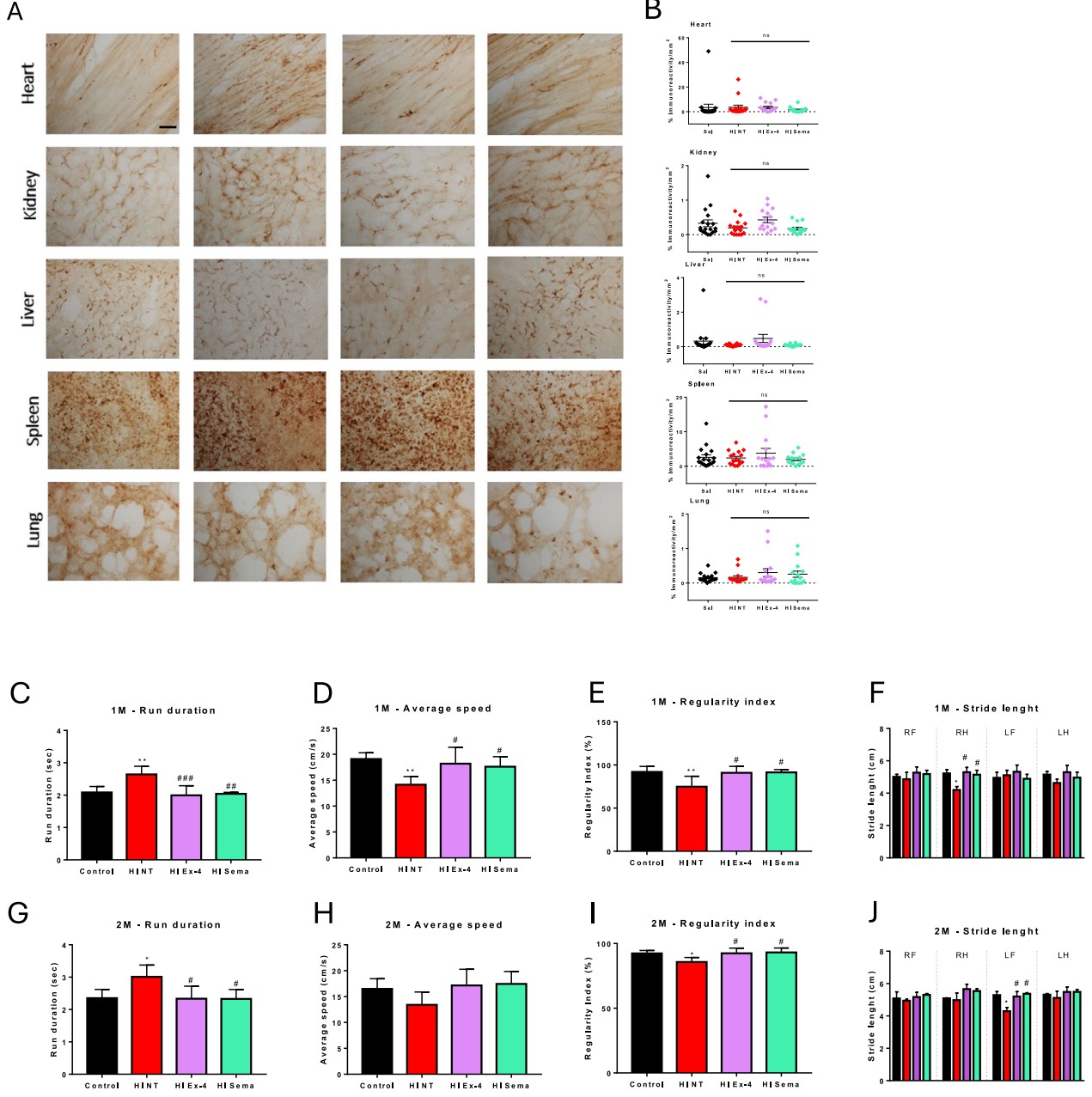

**Figure EV4.  Complementary results for long-term in vivo study post HI.**

(A, B) No significant microglia activation or infiltration was observed or measured in sections from organ harvested from saline controls (Sal, $n = 18$), HI non-treated (HINT, $n = 16$), HI treated with Exendin-4 (Ex-4, $n = 15$) and HI treated with Semaglutide animals (Sema, $n = 14$). Scale bar: 60 μm. Locomotor functions were evaluated at 1 and 2 months post-HI with Catwalk and revealed significant exendin-4 and semaglutide induced improvements at 4 weeks (C–F) and 8 weeks (G–J) of age compared with the age-matched hypoxic-ischaemic group for various parameters: (C, G) run duration, (D, H) average speed, (E, I) regularity index, and (F, J) stride length (RF = right front, RH = right hind, LF = left front, LH = left hind) in saline controls (Control, 1 M $n = 18$, 2 M $n = 15$), HI non-treated (HINT, 1 M $n = 16$, 2 M $n = 13$), HI treated with Exendin-4 (Ex-4, 1 M $n = 14$, 2 M $n = 12$) and HI treated with Semaglutide (Sema, 1 M $n = 21$, 2 M $n = 22$). Data information: Each n represents an individual mouse. Error bars indicate mean ± SD. Statistical analysis performed using an ordinary one-way analysis of variance (ANOVA) with a Dunnett's multiple comparisons test. * or #, $p < 0.05$; ** or ##, $p < 0.01$; *** or ###, $p < 0.001$; **** or ####, $p < 0.0001$. * to compare with control and # to compare with HI non-treated (NT) group.

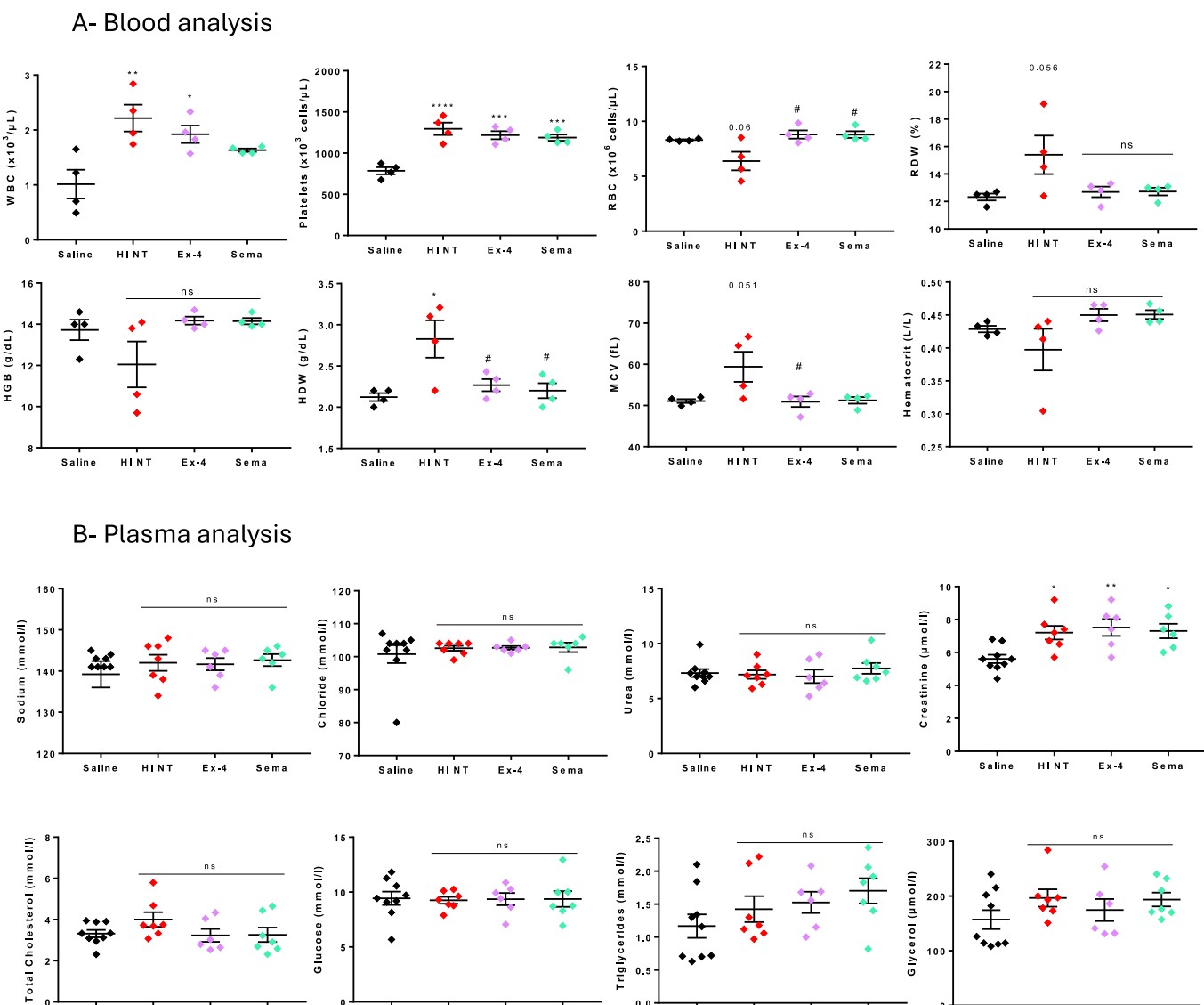

## A- Blood analysis

## B- Plasma analysis

**Figure EV5.  Blood and plasma analysis 8 weeks post-insult.**

(**A**) Blood analysis from the long-term 8 weeks study showed HI resulted in a significant increase in WBCs, platelets and haemoglobin distribution widths (HDWs) and a trend towards decreased RBCs, haemoglobin (HGB), increased red cell distribution widths (RDWs) and MCVs. Treatment with GLP1-R agonists improved the RBC, RDW, HGB, HDW and MCV counts (*n* = 4 per experimental group). (**B**) Analysis of plasma parameters (sodium, chloride, urea, creatinine, total cholesterol, glucose, triglycerides, glycerol levels) in saline controls (Saline, *n* = 9), saline-treated HI animals (HINT, *n* = 7), HI treated with Exendin-4 (Ex-4, *n* = 6) and HI treated with Semaglutide (Sema, *n* = 7), which showed no significant difference between groups except for a statistically significant increase in creatinine in all HI groups. Data information: Each n represents an individual mouse. Error bars indicate mean ± SD. Statistical analysis performed using an ordinary one-way analysis of variance (ANOVA) with a Dunnett's multiple comparisons test. * or #, *p* < 0.05; ** or ##, *p* < 0.01; *** or ###, *p* < 0.001; **** or ####, *p* < 0.0001. * to compare with control and # to compare with HI non-treated (NT) group.

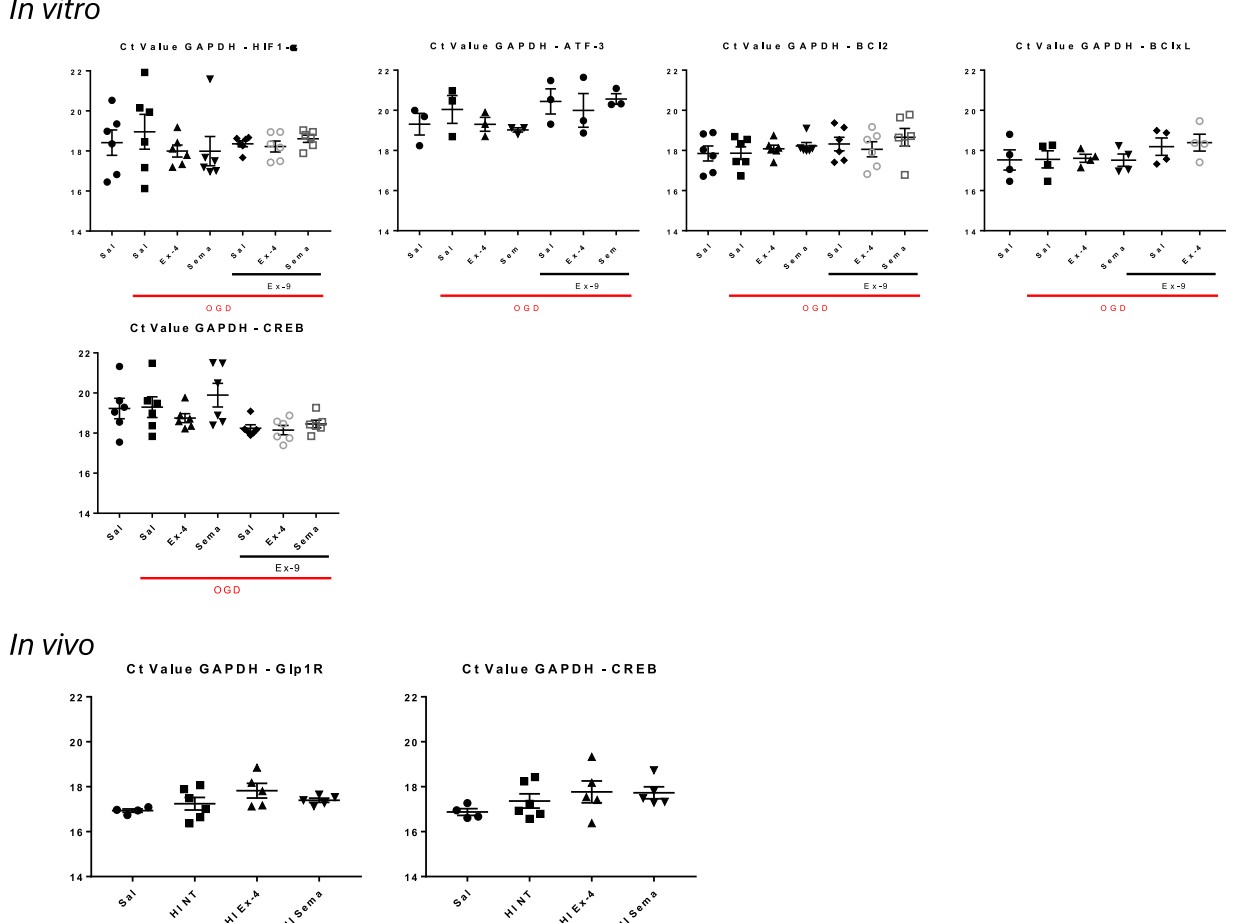

**Figure EV6. All qPCR control ct values are shown from the study.**

The GAPDH Ct value for various markers investigated by qPCR in the in vitro neuronal cells exposed to OGD following treatment with exendin-4 or semaglutide, and in combination with exendin-9: HIF-1-α ($n = 6$ for all experimental group), ATF-3 ($n = 3$ for all experimental groups), Bcl-2 ($n = 6$ for all experimental groups), Bcl-xL ($n = 4$ for all experimental groups) and CREB ($n = 6$ for all experimental groups). In addition, the Ct values for qPCR into Glp1R and CREB in the HIE model are also shown(Sal $n = 4$, HINT $n = 6$, HI Ex-4 $n = 5$ and HI Sema $n = 5$). Data information: error bars indicate mean ± SEM. Each $n$ represents an individual sample.

