## [Peer Review File · EMBO Molecular Medicine]

Diabetes drugs activate neuroprotective pathways in neonatal hypoxic-ischemic encephalopathy models

Laura Poupon-Bejuit, Amy Geard, Nathan Millicheap, Eridan Rocha Ferreira, Henrik Hagberg, Claire Thornton, and Ahad Rahim

Corresponding author: Ahad Rahim (a.rahim@ucl.ac.uk)

Review Timeline:

Submission Date:	6th Jun 22
Editorial Decision:	22nd Jun 22
Revision Received:	23rd Mar 24
Editorial Decision:	4th Apr 24
Revision Received:	30th Apr 24
Accepted:	7th May 24

Editor: Zeljko Durdevic

Transaction Report:

22nd Jun 2022

Dear Dr. Rahim,

Thank you for the submission of your manuscript to EMBO Molecular Medicine. We have now received feedback from the three reviewers who agreed to evaluate your manuscript. As you will see from the reports below, the referees acknowledge the interest of the study but also raise important concerns that should be addressed in a major revision.

We would welcome the submission of a revised version within three months for further consideration. Please let us know if you require longer to complete the revision.

I look forward to receiving your revised manuscript.

Yours sincerely,

Zeljko Durdevic

We require:

- 1) A .docx formatted version of the manuscript text (including legends for main figures, EV figures and tables). Please make sure that the changes are highlighted to be clearly visible.
- 2) Individual production quality figure files as .eps, .tif, .jpg (one file per figure). For guidance, download the 'Figure Guide PDF': (<https://www.embopress.org/page/journal/17574684/authorguide#figureformat>).
- 3) A .docx formatted letter INCLUDING the reviewers' reports and your detailed point-by-point responses to their comments. As part of the EMBO Press transparent editorial process, the point-by-point response is part of the Review Process File (RPF), which will be published alongside your paper.
- 4) A complete author checklist, which you can download from our author guidelines (<https://www.embopress.org/page/journal/17574684/authorguide#submissionofrevisions>). Please insert information in the checklist that is also reflected in the manuscript. The completed author checklist will also be part of the RPF.
- 5) Please note that all corresponding authors are required to supply an ORCID ID for their name upon submission of a revised manuscript.
- 6) It is mandatory to include a 'Data Availability' section after the Materials and Methods. Before submitting your revision, primary datasets produced in this study need to be deposited in an appropriate public database, and the accession numbers and

database listed under 'Data Availability'. Please remember to provide a reviewer password if the datasets are not yet public (see <https://www.embopress.org/page/journal/17574684/authorguide#dataavailability>).

8) We would also encourage you to include the source data for figure panels that show essential data. Numerical data should be provided as individual .xls or .csv files (including a tab describing the data). For blots or microscopy, uncropped images should be submitted (using a zip archive if multiple images need to be supplied for one panel). Additional information on source data and instruction on how to label the files are available at

9) Our journal encourages inclusion of *data citations in the reference list* to directly cite datasets that were re-used and obtained from public databases. Data citations in the article text are distinct from normal bibliographical citations and should directly link to the database records from which the data can be accessed. In the main text, data citations are formatted as follows: "Data ref: Smith et al, 2001" or "Data ref: NCBI Sequence Read Archive PRJNA342805, 2017". In the Reference list, data citations must be labeled with "[DATASET]". A data reference must provide the database name, accession number/identifiers and a resolvable link to the landing page from which the data can be accessed at the end of the reference. Further instructions are available at

13) Author contributions: the contribution of every author must be detailed in a separate section (before the acknowledgments).

14) A Conflict of Interest statement should be provided in the main text.

***** Reviewer's comments *****

Referee #1 (Comments on Novelty/Model System for Author):

The technical quality is medium because no information is provided on temperature control after HI. Small rodents consistently become hypothermia after HI, and this leads to confounding neuroprotection and must alter intracellular mechanism.

the medical impact is medium because treatment is started immediately after HI. This is not possible in any clinical trial.

the novelty is good in that it extends previous findings by the team, to include mechanisms, a much longer acting agent, and much longer-term outcomes. At the same time, they have already established that this class of agents have significant neuroprotective effects.

Referee #1 (Remarks for Author):

This team have recently shown that exendin-4 offers potent short-term neuroprotection protective after HI in neonatal mice. In this study they extend their original report to examine the specific mechanism of protection in vitro, demonstrate reversal with a specific antagonist and so exclude non receptor mediated effects, and confirm roughly comparable neuroprotection with the long-acting derivative semaglutide at much later times.

The results are scientifically very interesting, and greatly strengthen the case for future clinical translation.

The in vivo studies have a number of important strengths, including detailed assessment of adverse effects, a careful dose response, prolonged followup to either 7 or 60 days to confirm durable protection, with a complementary mixture of histology and neurobehavioral outcomes. The original studies with exendin-4 only examined 48h recovery, so this is critical additional information.

The finding of mild, treatable fall in blood glucose with semaglutide confirms previous theoretical concerns and will be an important consideration for future human translation.

The data presentation is excellent.

There are a few methodological limitations that must be addressed.

First, treatment was begun immediately after HI. This is clinically impossible, and will substantially augment the relative benefit. This must be frankly acknowledged as a major limitation.

Second, no information is provided on temperature control after HI. The environmental temperature is controlled to 37C during HI.

In neonatal animals spontaneous post HI hypothermia always occurs unless it is actively prevented, and is highly protective, and suppresses many intracellular pathways.

e.g. see: Rectal temperature in the first five hours after hypoxia-ischemia critically affects neuropathological outcomes in neonatal rats. Wood T, Hobbs C, Falck M, Brun AC, Løberg EM, Thoresen M. *Pediatr Res*. 2018 Feb;83(2):536-544. doi: 10.1038/pr.2017.51.

Further, temperature changes after HI can and have previously confounded drug effects. This was a major factor in the well known failure of glutaminergic antagonists to translate.

e.g. see this critical review:

Incomplete assessment of experimental cytoprotectants in rodent ischemia studies. DeBow SB, Clark DL, MacLellan CL, Colbourne F. *Can J Neurol Sci*. 2003 Nov;30(4):368-74. doi: 10.1017/s0317167100003097.

And more recently:

Klahr AC, et al *Ther Hypothermia Temp Manag.* 2017 Mar;7(1):42-49.

In view of this, please provide both rectal temperature measurements and details of environmental control after HI. Typically animal measurements are done using sentinel animals that are otherwise identically prepared.

The discussion is reasonable, but could be more critical in several aspects, particularly about the association of neuroprotection and inflammation.

The GLP1-R agonists were associated with both reduced injury and reduced microglial and astrocyte induction and reduced IL induction. These changes seem to be completely proportional to the reduced area and severity of brain infarction. No evidence is presented that suggests any primary effect on inflammation. More; we know that HI is associated with peripheral mobilisation of inflammatory cells, and yet there was no effect of the agonists on peripheral WCCs etc. On balance these data suggest that reduced inflammation was secondary to neuroprotection, likely mediated through reduced programmed cell death as illustrated in figure 8. This should be discussed frankly.

The discussion of hypoglycemia really doesn't explain why it was associated with semaglutide and not exendin-4. Were plasma insulin levels different between the GLP agonists?

Minor points

The abstract must specify the age of the mice, how HI was induced, and the drug protocols, particularly the immediate start of starting treatment.

Why were 2-month old mice given semaglutide? Viz "Postnatal Day 10 (P10) mice or 2-month-old mice underwent one injection of semaglutide" I cannot find results of this study but it may be embedded in other data.

A very minor question; there is a trend to greater residual infarct volumes after semaglutide at day 60. It isn't significant, but I submit that this study is not powered to compare the treatment groups. The trend is large enough that it would be materially important if confirmed in a larger, better powered study ((mean 1.09mm³ after exendin, vs 2.56mm³ after semaglutide vs 8.56mm³ after vehicle-HI). Would the authors perhaps like to speculate whether this might reflect longer maintenance of drug levels over time with the protocol repeated exendin-4, or some other drug specific?

Trivia.

The MS is written clearly but there is ambiguous phrasing in places and fragmented presentation of key methods. Careful revision is needed throughout. Some examples follow:

Please do not use "the HIE mouse model". There are many models in mice, and the word model provides no information. At first use specify the age and paradigm, then after that HI in neonatal mice. For example, "HI was induced by unilateral carotid occlusion in P10 CD1 mice, and either saline, exendin-4 or semaglutide were given immediately after HI". Please remove the word model throughout.

What is the significance if any of using CD1 mice?

"latent period of at least 6h," correct at least, to "up to"

Ibid. encephalopathy is present in the latent phase.

"an effectiveness of approximately 50% in moderate HI cases" here I think the authors mean nearly 50% of infants with moderate to severe HIE die or survive with disability. The benefit is greater in moderate HIE, e.g. see. *BMJ.* 2010 Feb 9;340:c363. doi: 10.1136/bmj.c363.

"the HI insult" all that is needed is HI. "Insult" per se means nothing
A mixture of US and UK spelling is used in different places. Completely trivial sorry.

HIF-1 α vs HIF1- α ?
hypoxia-ischemia should be abbreviated consistently

Methods.

Please describe the in vivo experiments in one place. At moment the methods are spread out in an illogical way so that it is difficult for the reader to quickly understand the study; the reader would want to know what type of animals, their age, the method of inducing HI, and when the agents are given. It took repeated searching for this respondent to understand the protocol. The authors may wish to consider adding a small flow chart to illustrate the protocols?

"The infarct volume was calculated as described previously." Given that it does not take much space, it would be better to describe it here, and confirm that you used the indirect method to avoid artifacts due to swelling and atrophy of the infarct over time? Space can easily be save elsewhere

Results:

The tense of data reporting is inconsistent. Most often results are reported in the past tense since the study has already occurred?

What does "after damage" mean here? Usually damage refers to the ultimate extent of cell or tissue loss.

Expression.... were? Possibly should be was?

P values. The number of significant figures is inconsistent and in places may represent false precision? I suggest 3 significant figures, but of course this is up to journal style.

Under "GLP1-R agonists ameliorate HI-induced cellular-mediated neuroinflammation in vivo". Why are adverse events embedded in this para? It would make more sense to create a new section and make it easier for the reader to find these results. Note the same issue for the 60 day recovery results.

What does "and were normalised to control measurements." mean? Is this a statistical adjustment?

Should exendin-4 be capitalized?

Referee #2 (Comments on Novelty/Model System for Author):

Models are adequate although the studies in OGD could be done in vivo.

Referee #2 (Remarks for Author):

The aim of the study is to demonstrate that both exendin-4 and semaglutide, GLP1-R agonists used for treatment of diabetes, can improve neuropathology, survival and neurological deficits after neonatal hypoxic-ischemic encephalopathy (HIE) in mice via the PI3/AKT and CREB signaling pathways.

This study showed that in response to GLP1-R activation, PI3K/AKT pathway and cAMP levels were increased in neonatal HIE mouse model and in an in vitro OGD model. Authors showed that GLP1-R agonists were able to attenuate apoptosis, inflammation and oxidative stress while blocking the receptor reversed those effects in vitro OGD model. Furthermore, authors compared the efficacy of exendin-4 versus semaglutide and demonstrated that semaglutide was able to ameliorate HIE damage after a single injection compared to exendin-4 which required four doses. In summary, this study provided a basis to decipher how the GLP1-R agonists protect the brain in vitro and in a mouse model of HIE.

Strengths:

- Translational significance of this study is high as current treatment modalities for HIE are lacking, therefore, investigating clinically approved drugs and routes of administration are important and necessary. If proven effective, it can easily be translated to clinics.
- The mechanism of GLP-1R agonist induced neuroprotection has not been investigated in HIE.
- Both in vivo and vitro are used. Both short-term and long-term effects were evaluated in HIE model. Multiple organs were examined and blood cells were analyzed.

Weaknesses:

- Lack of conceptual novelty. The authors have already published exendin-4's neuroprotective and anti-inflammatory properties after neonatal HIE. In this study, they are just adding another GLP1-R agonist, semaglutide, and comparing it to exendin-4. Furthermore, there are already pre-clinical studies that have demonstrated the neuroprotective effects of semaglutide in models of ischemic stroke.
- The authors state that they want to demonstrate the mechanism via which both GLP1-R agonists exert their neuroprotective properties. However, the mechanism studies are poorly designed. GLP1-R antagonist was only used in in vitro studies. No mechanism studies were done in vivo making it a descriptive study. Furthermore, PI3/AKT and CREB signaling pathway has been extensively studied which reduces novelty of the study.
- The authors have designed superficial experiments measuring various outcomes like apoptosis, inflammation, oxidative stress and astrogliosis. It will be more beneficial if authors designed in depth studies focusing on one or two outcomes.
- It is not clear how best dose was determined for the two agonists used. It appears the authors determined the dose based on cAMP levels. Authors need to perform Nissl staining to show percent infarcted area and show the progression of infarction over 4-5 slices per brain.
- It is not clear as to what is the purpose of using in vitro OGD model. All experiments can be conducted in vivo.
- Immunofluorescent images are poor quality and do not add any significant data.
- The stability of the model is questionable as about half of the animals presented with no infarction. A stable reproducible model needs to be presented before treatments are tested.
- It appears that in supplementary Figure 5 there are no significant differences in inflammatory or apoptotic factors seen from PCR data. Authors need to add western blots to show apoptotic and inflammatory markers with and without treatment.
- Specify if statistical methods were employed to predetermine the sample size and include a description of sample size calculation.
- Describe procedures for prevention of bias as well as how were animals randomized.

- The full name of GLP should be spelled out at its first appearance in the abstract.
- The rationale of using Ex-9 should be provided before the results.
- The viewed location in the brain should be indicated. A scale needs to be added into each image.

Referee #3 (Comments on Novelty/Model System for Author):

They used ARRIVE and both in vitro and in vivo systems.

Referee #3 (Remarks for Author):

The authors nicely demonstrate that GLP1-R agonists can achieve neuroprotection in a variety of ways. They show enhance activity of the PI3K/AKT pathway and increased levels of cAMP in response to activation. That show that semaglutide is able to lessen damage from a single injection and exendin-4 requires 4 doses. They show histological as well as behavioral measures to support their conclusions. They also show reduction of ATF3 in vivo and in vitro. They also show a reduction in HIF1a. Since HIF1a has been shown to be neuroprotective, perhaps some speculation on this reduction could be mentioned in the discussion. Is it because the tissue is no longer hypoxic?

Mortality should be shown. Also are there sex differences in the responses?

Response to Reviewer's Comments

We thank the Reviewers for their thorough evaluation of this manuscript. Below we respond point-by-point to each comment and highlight when appropriate where edits have been made in the manuscript. The Authors have no doubt that the manuscript has benefitted greatly through this review process and the study has been significantly enhanced.

Referee #1

Referee #1 (Comments on Novelty/Model System for Author):

Referee: The technical quality is medium because no information is provided on temperature control after HI. Small rodents consistently become hypothermia after HI, and this leads to confounding neuroprotection and must alter intracellular mechanism.

Authors: We thank the referee for this comment. We have responded to this point below.

Referee: The medical impact is medium because treatment is started immediately after HI. This is not possible in any clinical trial.

Authors: We also respond to this point below in more detail.

Referee: The novelty is good in that it extends previous findings by the team, to include mechanisms, a much longer acting agent, and much longer-term outcomes. At the same time, they have already established that this class of agents have significant neuroprotective effects.

Authors: We thank the Referee for highlighting that this study significantly builds on our previous studies through shedding light on mechanisms, using longer acting agents and longer-term outcomes. This information will have a significant impact on the various crucial decision-making processes required for clinical translation of this work in terms of safety and therapeutic efficacy.

Referee #1 (Remarks for Author):

Referee: This team have recently shown that exendin-4 offers potent short-term neuroprotection protective after HI in neonatal mice. In this study they extend their original report to examine the specific mechanism of protection in vitro, demonstrate reversal with a specific antagonist and so exclude non receptor mediated effects, and confirm roughly comparable neuroprotection with the long-acting derivative semaglutide at much later times.

The results are scientifically very interesting, and greatly strengthen the case for future clinical translation.

Authors: We share the Referee's view in the importance of the mechanistic aspect of the study in strengthening the case for clinical translation. We feel that this data is crucial in guiding the translational pathway.

Referee: The in vivo studies have a number of important strengths, including detailed assessment of adverse effects, a careful dose response, prolonged followup to either 7 or 60 days to confirm durable protection, with a complementary mixture of histology and neurobehavioral outcomes. The original studies with exendin-4 only examined 48h recovery, so this is critical additional information. The finding of mild, treatable fall in blood glucose with semaglutide confirms previous theoretical concerns and will be an important consideration for future human translation.

Authors: We fully agree with the Referee's view that the *in vivo* data in this study will support clinical translation but also inform us of potential concerns, as highlighted by the fall in blood glucose observed when using semaglutide.

Referee: The data presentation is excellent.

Authors: The Authors appreciate the Referee's support.

Referee: There are a few methodological limitations that must be addressed. First, treatment was begun immediately after HI. This is clinically impossible, and will substantially augment the relative benefit. This must be frankly acknowledged as a major limitation.

Authors: The Referee has raised an important point. We have now clearly stated in the Discussion section that we have administered the GLP1-R agonists immediately after HI injury and that this may not fully reflect what may happen in the clinic. However, we have previously demonstrated in the same mouse model that intervention delayed by 4 hours post HI injury still has a significant therapeutic effect (Rocha-Ferreira et al, Brain, 2018). The following text has been added to the Discussion on page 23:

'An important consideration is that in this study the GLP1-R agonists were administered immediately following HI injury. This may not fully reflect what may happen in a clinical scenario. However, we have previously demonstrated that the therapeutic window of opportunity using a GLP1-R agonist (exendin-4) can still be significantly effective in ameliorating HI injury in the brain even when administered at 4 hours post injury.'

Referee: Second, no information is provided on temperature control after HI. The environmental temperature is controlled to 37C during HI. In neonatal animals spontaneous post HI hypothermia always occurs unless it is actively prevented, and is highly protective, and suppresses many intracellular pathways. e.g. see: Rectal temperature in the first five hours after hypoxia-ischemia critically affects neuropathological outcomes in neonatal rats. Wood T, Hobbs C, Falck M, Brun AC, Løberg EM, Thoresen M. *Pediatr Res.* 2018 Feb;83(2):536-544. doi: 10.1038/pr.2017.51.

Further, temperature changes after HI can and have previously confounded drug effects. This was a major factor in the well known failure of glutaminergic antagonists to translate.

e.g. see this critical review:

Incomplete assessment of experimental cytoprotectants in rodent ischemia studies. DeBow SB, Clark DL, MacLellan CL, Colbourne F. *Can J Neurol Sci.* 2003 Nov;30(4):368-74. doi: 10.1017/s0317167100003097.

And more recently:

Klahr AC, et al *Ther Hypothermia Temp Manag.* 2017 Mar;7(1):42-49.

In view of this, please provide both rectal temperature measurements and details of environmental control after HI. Typically animal measurements are done using sentinel animals that are otherwise identically prepared.

Authors: We appreciate the Referee's point about spontaneous post-HI hypothermia. It is an important point and all Authors have significant experience in working with perinatal animals and their methodologies and environmental factors are designed to prevent a drop in core body temperature. We have now added a sentence to the methodology on page 7 highlighting this and also cited the very useful reference that the Referee has provided (Klahr et al., 2017):

'Body temperature was maintained during surgery and incubations to prevent hypothermia that can confound therapeutic readouts and the effect of drugs (Klahr et al., 2017). This was achieved using heating mats, heated incubators and prompt returning of pups to the dam'.

We do not routinely monitor core body temperature using an anal probe since the NC3Rs does not encourage its use. We have done this in the past when inducing hypothermia since there is a compelling reason to monitor the drop in core body temperature using an anal probe and shown that we are able to regulate temperature effectively (Roche-Ferreira et al, 2018, Brain). We have now added more clarity in the text on how we try to maintain body temperature and details on the external temperature. This includes the insertion of the following text in the Materials and Methods Section on page 7:

'The body temperature was maintained using a heating mat' and 'mice recovered in an incubator at 37°C and were returned to the dam for 1h'.

We have already included text on how the pups were *'then placed in a hypoxia chamber and exposed to humidified 8% oxygen/92% nitrogen (3L/min) at 37°C for 20 min, resulting in moderate to severe brain damage 11,28'*. We have also added the following text after the description of the pups in the hypoxia chamber:

'The pups were then immediately returned to the dam'

These measures to prevent a drop in core body temperature are constant across all experimental cohorts and Figures 4 and 5 demonstrate that a significant injury is measured in the brain through formation of an infarct and in inflammatory response, respectively, compared to naïve controls.

Referee: The discussion is reasonable, but could be more critical in several aspects, particularly about the association of neuroprotection and inflammation.

The GLP1-R agonists were associated with both reduced injury and reduced microglial and astrocyte induction and reduced IL induction. These changes seem to be completely proportional to the reduced area and severity of brain infarction. No evidence is presented that suggests any primary effect on inflammation. More; we know that HI is associated with peripheral mobilisation of inflammatory cells, and yet there was no effect of the agonists on peripheral WCCs etc. On balance these data suggest that reduced inflammation was secondary to neuroprotection, likely mediated through reduced programmed cell death as illustrated in figure 8. This should be discussed frankly.

Authors: The referee raises an interesting point. We have now included the below text into the Discussion on page 22-23 that addresses this while also being careful not to over-interpret the data and acknowledging the fact that more work needs to be done to definitively dissect the sequence of events.

'It remains unclear whether the reduced inflammation is a direct consequence of the drugs acting primarily on glial cells, or whether it is a consequence of neuroprotection, or both. We have previously shown that GLP1-R is present on neurons and astrocytes at this early stage of development.¹⁸ Although the reduction in infarct size and programmed cell death may suggest that the amelioration in inflammatory response is a secondary response, the precise sequence of events requires further investigation'.

Referee: The discussion of hypoglycemia really doesn't explain why it was associated with semaglutide and not exendin-4. Were plasma insulin levels different between the GLP agonists?

Authors: The reason why semaglutide induces hypoglycaemia at high doses and exendin-4 does not is an interesting observation and requires more studies. Although we measured and report on blood

glucose levels (Supplementary Figure 2E), we did not measure plasma insulin. In the absence of data that may further cast light upon this observation, we have been careful not to over-interpret and speculate. However, we absolutely recognise the need to further dissect hypoglycaemia in pups using high doses of semaglutide and have acknowledged this in the Discussion on page 21-22 with the inclusion of the following text:

'The reason for why semaglutide induces hypoglycaemia and the equivalent dose of exendin-4 is not fully understood and requires further investigation'.

Minor points

Referee: The abstract must specify the age of the mice, how HI was induced, and the drug protocols, particularly the immediate start of starting treatment.

Authors: We have now added the requested details, while trying to adhere to the limited word count of the abstract, through the inclusion of the following text:

'In this study, we demonstrate that post-natal day 10 mice with surgically induced hypoxic-ischemic brain injury can have significantly improved neurological outcomes when either exendin-4 or semaglutide are systemically administered immediately after injury. Measurements at short- and long-term time points show improved neuropathology, survival and locomotor function'.

Referee: Why were 2-month old mice given semaglutide? Viz "Postnatal Day 10 (P10) mice or 2-month-old mice underwent one injection of semaglutide" I cannot find results of this study but it may be embedded in other data.

Authors: We apologies to the Referee for this typo and thank them for highlighting this error. We have corrected this by deleting the mention of 2 month old mice being administered on page 10. Indeed, 2 month old mice were examined post-treatment at P10, but not administered at 2 months.

Referee: A very minor question; there is a trend to greater residual infarct volumes after semaglutide at day 60. It isn't significant, but I submit that this study is not powered to compare the treatment groups. The trend is large enough that it would be materially important if confirmed in a larger, better powered study ((mean 1.09mm³ after exendin, vs 2.56mm³ after semaglutide vs 8.56mm³ after vehicle-HI). Would the authors perhaps like to speculate whether this might reflect longer maintenance of drug levels over time with the protocol repeated exendin-4, or some other drug specific?

Authors: We have now removed the infarct volume measurements to address the Reviewer's comments further below on volume measurements and artifacts.

Trivia.

Referee: The MS is written clearly but there is ambiguous phrasing in places and fragmented presentation of key methods. Careful revision is needed throughout. Some examples follow: Please do not use "the HIE mouse model". There are many models in mice, and the word model provides no information. At first use specify the age and paradigm, then after that HI in neonatal mice. For example, "HI was induced by unilateral carotid occlusion in P10 CD1 mice, and either saline, exendin-4 or semaglutide were given immediately after HI". Please remove the word model throughout.

Authors: We have removed referencing "HIE mouse model" or reference to 'model' throughout the manuscript including the abstract on page 1, the introduction on page 3 and 4, the Materials and Methods on page 4 and 6, the Results section on page 12, 13, 14, 15, 19, the Discussion page 20, 21,

22, the Conclusion page 23 and Figure 2 legend page 29. The exception is the title where 'model' remains to reflect the *in vitro* and *in vivo* approaches rather than referring directly to HI injury in the P10 pups.

Referee: What is the significance if any of using CD1 mice?

Authors: CD1 mice have large litter sizes and the mothers have excellent maternal instincts. This is an advantage for the care of pups that are separated from mothers for an extended period of time. The following text has been added to the Materials and Methods section on page 7 to describe this: *'The CD1 strain was used due to their large litter sizes and excellent maternal instincts. This is advantageous where pups are separated from their mothers for extended periods of time.'*

Referee: "latent period of at least 6h," correct at least, to "up to"

Authors: This has now been corrected on page 3.

Referee: Ibid. encephalopathy is present in the latent phase.

Authors: This has been corrected on page 3.

Referee: "an effectiveness of approximately 50% in moderate HI cases" here I think the authors mean nearly 50% of infants with moderate to severe HIE die or survive with disability. The benefit is greater in moderate HIE, e.g. see. BMJ. 2010 Feb 9;340:c363. doi: 10.1136/bmj.c363.

Authors: We have now corrected the language in the Introduction on page 3 as per the Referee's suggestion and used the citation that they have kindly provided.

Referee: "the HI insult" all that is needed is HI. "Insult" per se means nothing

Authors: 'HI insult' has now been removed throughout the manuscript.

Referee: A mixture of US and UK spelling is used in different places. Completely trivial sorry.

Authors: We have been through the manuscript to try and maintain to UK spelling.

Referee: HIF-1 α vs HIF1- α ?

Authors: We thank the Referee for highlighting this. HIF-1 α is now used consistently throughout the manuscript.

Referee: hypoxia-ischemia should be abbreviated consistently

Authors: We have now consistently abbreviated hypoxia-ischemia as 'HI' throughout the manuscript with the exception of the first time it is used and in the sub-heading on page 7.

Referee: Methods.

Please describe the *in vivo* experiments in one place. At moment the methods are spread out in an illogical way so that it is difficult for the reader to quickly understand the study; the reader would want to know what type of animals, their age, the method of inducing HI, and when the agents are given. It took repeated searching for this respondent to understand the protocol. The authors may wish to consider adding a small flow chart to illustrate the protocols?

Authors: We have now expanded the *in vivo* section with more detail and in one place on page 7 of the Materials and Methods section. We have also added more information on the treatment regimen. The text now reads as follows on pages 7:

'Surgically induced hypoxia-ischemia in P10 CD1 mice and treatments

All experiments involved litters of postnatal day 10 CD1 mice (P10) bred in-house. The CD1 strain was used due to their large litter sizes and excellent maternal instincts. This is advantageous where pups are separated from their mothers for extended periods of time. The surgical procedures were performed as previously described.^{18,27,28} Body temperature was maintained during surgery and incubations to prevent hypothermia that can confound therapeutic readouts and the effect of drugs (Klabr et al., 2017). This was achieved using heating mats, heated incubators and prompt returning of pups to the dam. Briefly, male and female P10 mice were anesthetized with isoflurane (5% induction and 1.5% maintenance). The left common carotid artery was permanently occluded with 8/0 polypropylene suture and the wound closed with tissue glue. The body temperature was maintained using a heating mat. The mice recovered in an incubator at 36°C and were returned to the dam for 1h. The pups were then placed in a hypoxia chamber and exposed to humidified 8% oxygen/92% nitrogen (3L/min) at 37°C for 20 min, resulting in moderate to severe brain damage^{11,28} The pups were then immediately returned to the dam. The P10 rodent induction of HI through unilateral occlusion of the carotid presents phenotypical similarities to the grey and white matter injury observed in humans, i.e., tissue loss, cell-death, microglial-mediated immune response, and astrogliosis as well as alteration in neurobehavioral performance.²⁹ All treatments were administered immediately after HI via intraperitoneal administration. Four doses of exendin-4 were administered in total, one every 12 hours (each dose of 0.5µg/g). Animals were randomized for short term (ST) and long term (LT) studies to: (i) saline (n =16 ST; n=18 LT); (ii) hypoxic-ischemic group with saline treatment (n =18 ST; n=19 LT); (iii) hypoxic-ischemic group with four high doses of exendin-4 (0.5µg/g) (ip) administered every 12 h, starting immediately after HI (n =18 ST; n=18 LT); (iv) hypoxic-ischemic group with a single dose of semaglutide (0.25µg/g) (ip) administered immediately after HI (n =17 ST; n=18 LT). Treated animals were culled at either 48h post HI for ST studies or 60 days post HI for LT studies. The study design is illustrated in Supplementary Figure 1.

We have also taken the Referees very good suggestion of adding an illustration of the study design which can now be found as Supplementary Figure 1.

Referee: "The infarct volume was calculated as described previously." Given that it does not take much space, it would be better to describe it here, and confirm that you used the indirect method to avoid artifacts due to swelling and atrophy of the infarct over time? Space can easily be saved elsewhere

Authors: We apologise for this. Since the completion of the study, the post-doc working on this project has left, moved abroad and started a family. We have spent considerable time trying to contact them to get clarity on the specifics of the methodology used to avoid artifacts and writing in the lab book is difficult to follow without their guidance. Unfortunately, we have not been able to make contact. As a result and to maintain scientific integrity, we decided to remove the infarct volume measures and rely on the macroscopic scores to measure infarct size and have retained the representative pictures of the brain sections showing damage. We hope the Reviewer understands our reason for making this decision.

Referee: Results:

The tense of data reporting is inconsistent. Most often results are reported in the past tense since the study has already occurred?

Authors: Thank you. We have been through the manuscript and addressed this.

Referee: What does "after damage" mean here? Usually damage refers to the ultimate extent of cell or tissue loss.

Authors: This has been corrected to '*Neuronal death was observed 2h after OGD by a significant decline in survival to 67% (Fig 2E) ($p < 0.0001$)*' on page 13 of the Results section.

Referee: Expression.... were? Possibly should be was?

Authors: This has now been changed on page 14 of the Results section to read '*Expression of the Bcl-2 family of proteins, which are essential mitochondrial apoptosis regulators, was investigated by qPCR in primary neuron cultures after OGD with or without GLP1-R agonist treatment*'.

Referee: P values. The number of significant figures is inconsistent and in places may represent false precision? I suggest 3 significant figures, but of course this is up to journal style.

Authors: Thank you, we have now standardised to 4 decimal places.

Referee: Under "GLP1-R agonists ameliorate HI-induced cellular-mediated neuroinflammation in vivo". Why are adverse events embedded in this para? It would make more sense to create a new section and make it easier for the reader to find these results. Note the same issue for the 60 day recovery results.

Authors: We have now created a new section called '*No adverse events associated to a single high dose administration of semaglutide in naïve P10 CD1 mice*' in the Results section on page 17. The text associated to this has now been transferred under this heading.

We have also created a new section on page 16 of the Results section '*GLP1-R agonists ameliorate longer-term HI-induced cellular-mediated neuroinflammation in vivo*'. This has now allowed the 60 day data to be separate from the 48h data and easier to follow.

Referee: What does "and were normalised to control measurements." mean? Is this a statistical adjustment?

Authors: We have changed the language to '*... and were comparable to control measurements*' on page 18 to make our meaning clearer and that it does not involve statistical adjustments of any kind.

Referee: Should exendin-4 be capitalized?

Authors: There is no consensus on this. Different publications have chosen to either capitalise or not. Our previous publication did not use capitals and so for consistency we have continued with this and standardised throughout the manuscript. We are very happy to conform to an Editorial decision.

Referee #2

(Comments on Novelty/Model System for Author):

Models are adequate although the studies in OGD could be done in vivo.

Referee #2 (Remarks for Author):

The aim of the study is to demonstrate that both exendin-4 and semaglutide, GLP1-R agonists used for treatment of diabetes, can improve neuropathology, survival and neurological deficits after neonatal hypoxic-ischemic encephalopathy (HIE) in mice via the PI3/AKT and CREB signaling

pathways.

This study showed that in response to GLP1-R activation, PI3K/AKT pathway and cAMP levels were increased in neonatal HIE mouse model and in an in vitro OGD model. Authors showed that GLP1-R agonists were able to attenuate apoptosis, inflammation and oxidative stress while blocking the receptor reversed those effects in vitro OGD model. Furthermore, authors compared the efficacy of exendin-4 versus semaglutide and demonstrated that semaglutide was able to ameliorate HIE damage after a single injection compared to exendin-4 which required four doses. In summary, this study provided a basis to decipher how the GLP1-R agonists protect the brain in vitro and in a mouse model of HIE.

Strengths:

Referee: Translational significance of this study is high as current treatment modalities for HIE are lacking, therefore, investigating clinically approved drugs and routes of administration are important and necessary. If proven effective, it can easily be translated to clinics.

Authors: We thank the referee for the thorough review of this manuscript and highlighting the translational importance of this study. We fully agree that GLP1-R agonists have significant potential for clinical use in HIE.

Referee: The mechanism of GLP-1R agonist induced neuroprotection has not been investigated in HIE.

Authors: We agree that this study does provide a mechanism of action of these GLP1-R agonists in the context of HIE. Understanding the mechanism of neuroprotection will allow us to make more informed decisions on how best to move this approach to the clinic. Furthermore, the hypoglycaemia observed using semaglutide also provides us with key information of risks associated to the approach and how we are best prepared to mitigate these.

Referee: Both in vivo and vitro are used. Both short-term and long-term effects were evaluated in HIE model. Multiple organs were examined and blood cells were analyzed.

Authors: We thank the Referee for highlighting the use of both in vitro and in vivo models to examine mechanisms behind the therapeutic effect of GLP1-R agonists in HIE.

Weaknesses:

Referee: Lack of conceptual novelty. The authors have already published exendin-4's neuroprotective and anti-inflammatory properties after neonatal HIE. In this study, they are just adding another GLP1-R agonist, semaglutide, and comparing it to exendin-4. Furthermore, there are already pre-clinical studies that have demonstrated the neuroprotective effects of semaglutide in models of ischemic stroke.

Authors: We have previously published that a GLP1-R agonist (exendin-4) has neuroprotective and anti-inflammatory effects in mice with HI (Roche-Ferreira et al, 2018, Brain). However, this study significantly moves the field forward through the following that has never been investigated until now in HIE; (i) We have furthered our understanding of whether a new generation of GLP1-R agonist (semaglutide) has any advantages over exendin-4 and any safety issues. This is a critical decision when planning for potential clinical translation and we have highlighted a potential issue regarding semaglutide but also how this can be mitigated e.g. hypoglycaemia and how this can be prevented through supplementation of glucose, (ii) Identifying the different mechanisms of action

involved in how GLP1-R agonists are able to protect neurons and the various cellular and gene expression responses that provide therapeutic benefit. Any information on how these drugs are having an effect will be a key component of supporting clinical translation and dialogue with regulators (iii) We show that the therapeutic effects of both exendin-4 and semaglutide are long-term and not just confined to a short 48h period after HI, and (iv) We demonstrate the adverse effects that HI has on locomotor function and behaviour using advanced techniques (e.g. Catwalk XT gait analysis) and how GLP1-R agonists have a significant therapeutic benefit in preventing this long-term. Taken together, this is large body of evidence supportive of clinical translation that has not been reported before.

Referee: The authors state that they want to demonstrate the mechanism via which both GLP1-R agonists exert their neuroprotective properties. However, the mechanism studies are poorly designed. GLP1-R antagonist was only used in *in vitro* studies. No mechanism studies were done *in vivo* making it a descriptive study. Furthermore, PI3/AKT and CREB signaling pathway has been extensively studied which reduces novelty of the study.

Authors: One of the scientific questions that we were attempting to address was whether the neuroprotective effect of exendin-4 and semaglutide was lost if the GLP1-R receptor was blocked using the antagonist (ex-9). The *in vitro* OGD model allows for a 'clean' readout in a single cell type (neurons) compared to an *in vivo* model where there are multiple cell types involved and convolutes the read out. In the *in vitro* OGD model, the use of the antagonist ex-9 clearly inhibits the effects of exendin-4 and semaglutide through reducing cell viability, the upregulation of pro-survival molecules and downregulation of apoptotic molecules, as shown in Figure 2. Having addressed the scientific question and reaching statistical significance, the impetus to conduct the same experiments in mice that have received HI was not compelling and non-compliant with NC3Rs guidelines. Figure 8 is entirely dedicated to addressing mechanisms *in vivo* including qPCR data of GLP1-R levels in response to treatment, pro-survival gene expression, protein levels and phosphorylation status of key molecules e.g. CREB, GSK3beta, PI3K and AKT.

Referee: The authors have designed superficial experiments measuring various outcomes like apoptosis, inflammation, oxidative stress and astrogliosis. It will be more beneficial if authors designed in depth studies focusing on one or two outcomes.

Authors: This study has provided important information of the various cellular responses that are activated when exendin-4 or semaglutide are administered following OGD or HI and are working together to provide neuroprotection e.g. upregulation of the GLP1-R receptor gene, reduced levels of HIF1-alpha, reduction in cellular inflammation, upregulation of CREB expression and its phosphorylation, increased phosphorylation of GSK3beta S9, PI3K and AKT, reduced levels of pro-apoptotic enzyme caspase 3, 8 and 9, amongst others molecules reported in Figures 2, 3, 5, 6, 8 and Supplementary Figure 2. However, we do agree with the Reviewer that there is significant scope to further deep dive into all these components, which we do plan to do in subsequent studies. We have acknowledged this in the Conclusion on page 23 through insertion of the following text:

'There is significant scope for more in-depth studies of any of the mechanistic outcomes that we have reported on'.

Referee: It is not clear how best dose was determined for the two agonists used. It appears the authors determined the dose based on cAMP levels. Authors need to perform Nissl staining to show percent infarcted area and show the progression of infarction over 4-5 slices per brain

Authors: We have described the dose, dosing regimen and the rationale behind it in the Results section on page 15 which reads as follows:

*'An effective dosing regimen for exendin-4 was determined in our previous study¹⁸ at 0.5 µg/g given every 12 h over a 48-h period and which also did not cause hypoglycaemia. We investigated glucose levels in P10 neonatal mice that received HI before and after administration with 0.5µg/g semaglutide at various timepoints and also via IP vs SC routes of administration (Supp Fig 2E). Statistically significant hypoglycaemia was detected at 30 min (**p= 0.0002 for IP), 2h (**p= 0.0003 for IP, *p = 0.018 for SC) and 24h (**p= 0.0003 for IP, *p = 0.0289 for SC) post-injection with semaglutide via the indicated routes. We also observed increased mortality. However, this was completely compensated for by oral supplementation with 2mg/g glucose (Supp Fig 2E). Thereafter, a dosing experiment was conducted to evaluate the optimal dose for semaglutide in the same neonatal HIE model (Supp Fig 2F) and brain protection was measured using a macroscopic scoring system to assess the size of the infarct. Here, the addition of oral glucose supplementation prevented the increased mortality observed in neonatal mice administered with the peptide following HI, and we observed that a single dose of semaglutide (0.25µg/g) shows the same therapeutic effect as 4 doses of exendin-4 (0.5 µg/g: Supp Fig 2F).'*

We have now removed the infarct volume measurements in response to Reviewer 1 comments but retain the macroscopic score of damage. However, we have retained the stained brain sections from each experimental group showing infarct in Figure 4.

Referee: It is not clear as to what is the purpose of using in vitro OGD model. All experiments can be conducted in vivo.

Authors: The complex mix of neural cells in the brain (e.g. neurons, astrocytes, microglia, ependymal cells) makes studying the mechanisms that are supporting neuroprotection and survival of neurons after HI complex and difficult. By using an in vitro primary neuronal cell culture approach, we are specifically measuring effects associated to OGD and how exendin-4 and semaglutide modulate these. It provides a 'cleaner' readout. We have added the following text into the Results section on page 12 to make this clearer.

'By using this in vitro primary neuronal cell approach, we were able to focus on neurons and mechanisms involved in neuroprotection without other neural cells to obfuscate the readouts'.

Referee: Immunofluorescent images are poor quality and do not add any significant data.

Authors: These have now been removed from the manuscript.

Referee: The stability of the model is questionable as about half of the animals presented with no infarction. A stable reproducible model needs to be presented before treatments are tested.

Authors: The Rice-Vannucci model of perinatal hypoxic-ischemic encephalopathy was first described in 1981 and is by far the most commonly used model to study pathophysiology, neuroprotection, neuroinflammation and new therapies for this condition. There are hundreds of papers using it as a research tool. It is generally accepted that there is considerable variation in infarct presence and size. In fact, one of the co-authors of this manuscript has studied this variability in depth using different strains of mice (Rocha-Ferreira et al., 2015, Neuroscience). However, this variability is taken into account using power calculations (described in more detail below in response to the Referee's comment on statistical methods). Indeed, the study is fully powered and statistical significance using the model is achieved throughout the manuscript in Figures 4-8. We have now added a description of the power calculations used under study design in the Materials and Methods section on page 5:

'Sample size was based on 5% significance with 80% power with one mouse representing an experimental unit. Effect size was estimated from our recent use of the animal model by Rocha-Ferreira et al.¹⁸ comparing HI (Veh) with HI (Treatment) groups and calculations performed using publicly available PS: Power and Sample Size Calculation v3.1.6 software'.

Referee: It appears that in supplementary Figure 5 there are no significant differences in inflammatory or apoptotic factors seen from PCR data. Authors need to add western blots to show apoptotic and inflammatory markers with and without treatment.

Authors: The data that was originally in Supplementary Fig 5 (now Supplementary Figure 6) shows the control values for all qPCR data conducted in this manuscript. We have presented this for transparency and there should not be any statistical difference between the samples, as the figures demonstrates in all qPCR reactions. This has now been made clearer in the text within the Results section on page 19 through the inclusion of the following text:

‘All qPCR control ct values are shown in Supp Fig 6’.

Referee: Specify if statistical methods were employed to predetermine the sample size and include a description of sample size calculation.

Authors: We have now included the following text in the Materials and Methods section under Study Design with the following text on page 5:

‘Sample size was based on 5% significance with 80% power with one mouse representing an experimental unit. Effect size was estimated from our recent use of the animal model by Rocha-Ferreira et al.¹⁸ comparing HI (Veh) with HI (Treatment) groups and calculations performed using publicly available PS: Power and Sample Size Calculation v3.1.6 software’.

Referee: Describe procedures for prevention of bias as well as how were animals randomized.

Authors: We have described this in the Materials and Methods section on page 4:

‘For the in vivo studies, animals were randomly allocated by one researcher to the control and treatment groups, with identifying marks for each experimental cohort. The next researcher was blinded to the identity of each group for behavioural analysis. The animals were housed together to minimize environmental differences and experimental bias’.

We have also now included a schematic of experimental design as Supplementary Figure 1 which also shows this.

Referee: The full name of GLP should be spelled out at its first appearance in the abstract.

Authors: We thanks the Referee for highlighting this. It has now been corrected.

Referee: The rationale of using Ex-9 should be provided before the results.

Authors: We have now included the reasons for including exendin-9 in the Materials and Methods section on page 6 through the inclusion of the following text:

‘Exendin-9 is an antagonist of the GLP1-R receptor and used in these to study to block the receptor and confirm inhibition of the exendin-4 and semaglutide binding’.

Referee #3

(Comments on Novelty/Model System for Author):

They used ARRIVE and both in vitro and in vivo systems.

(Remarks for Author):

Referee: The authors nicely demonstrate that GLP1-R agonists can achieve neuroprotection in a variety of ways. They show enhance activity of the PI3K/AKT pathway and increased levels of cAMP in response to activation. That show that semaglutide is able to lessen damage from a single injection and exendin-4 requires 4 doses. They show histological as well as behavioral measures to support their conclusions. They also show reduction of ATF3 in vivo and in vitro.

Authors: We thanks the Referee for their supportive comments and for highlighting that GLP1-R agonists do trigger a variety of neuroprotective responses in cells.

Referee: They also show a reduction in HIF1a. Since HIF1a has been shown to be neuroprotective, perhaps some speculation on this reduction could be mentioned in the discussion. Is it because the tissue is no longer hypoxic?

Authors: The Reviewer raises an interesting point. We have written about HIF-1- α on page 12 of the Results section with the following text:

HIF-1 α is a transcription factor that responds to changes in oxygen levels and has been found to be upregulated in brains after hypoxic and ischemic exposures.³³ Stabilised HIF-1- α can induce a variety of responses to hypoxia ranging from prosurvival to prodeath depending on the severity of the insult³⁴; in the neonatal brain, HIF-1- α has been shown to exert protective effects in neonatal HI.³⁵

Although, how either GLP1-R agonists are reducing the levels of HIF-1- α is unclear to us and needs further investigation. We have tried to be upfront about this and have now inserted into page 21 of the Discussion section:

'Although how HIF-1 α is reduced by the action of both GLP1-R agonists is unclear from our data'.

Referee: Mortality should be shown.

Authors: Thank you for highlighting this and we have now stated the mortality rate in the Materials and Methods section describing the surgically induced HI on page 7 with the inclusion of the following text:

'1% mortality was observed following this procedure'

Referee: Also are there sex differences in the responses?

Authors: No differences between sexes were noted in response to the treatments. This is consistent with our previous published work. The following text has now been inserted to the Results section on page 15.

No differences between male and female response to the treatments were noted and is consistent with our previous findings using exendin-4'.¹⁸

4th Apr 2024

Dear Prof. Rahim,

Thank you for the submission of your revised manuscript to EMBO Molecular Medicine. I am pleased to inform you that we will be able to accept your manuscript pending the following final amendments:

- 1) Please implement referee #1 suggestions.
- 2) Figures:
 - Please upload individual, high-resolution files in TIFF, EPS or PDF format for each main and EV figures. EV figure legends should be removed from the figure files and placed at the end of the manuscript file after main figure legends. Please provide detailed description of EV figures in their figure legends, as you have done in main figure legends. For more information on figure presentation please check "Author Guidelines".
<https://www.embopress.org/page/journal/17574684/authorguide#datapresentationformat>
 - We note that some panels are reused. Panels for Naïve and Saline spleen in EV Figure 3B are identical. Please check and clarify.
- 3) In the main manuscript file, please do the following:
 - Please address all comments suggested by our data editors listed below:
 - o Figure legends:
 1. Please note that a separate 'Data Information' section is required in the legends of figures 1a-2. b; 2b-i; 3a-b; 4b, e; 7a-b; 8b-c, e-f, h-i.
 2. Please define the annotated p values \$\$\$\$/\$\$\$/\$/\$/ ****/****/**/ #####/###/##/ in the legend of figure 2b-i; supplementary figures 2a-c, e-f; 4b-j; 5a-b; as appropriate.
 3. Please indicate the statistical test used for data analysis in the legends of supplementary figures 2a-c, e-f; 4b-j; 5a-b.
 4. Please note that in figures 2b-i; 3a-b; 4b, e; 5a-c; 7a-b; there is a mismatch between the annotated p values in the figure legend and the annotated p values in the figure file that should be corrected.
 5. Please note that information related to n is missing in the legends of figures 2b-i; 3a-b; 4e; 7a-b; 8b-c, e-f, h-i; supplementary figures 2a-c, e-f; 3c-d; 4b-j; 5a-b; 6.
 6. Although 'n' is provided, please describe the nature of entity for 'n' in the legends of figures 4b; 5a-c; 6a-c.
 7. Please note that the error bars are not defined in the legends of figures 1a-b; 2b-i; 3a-b; 4b, e; 5a-c; 6a-c; 7a-b; 8b-c, e-f, h-i; supplementary figures 2a-c, e-f; 3c-d; 4b-j; 5a-b; 6.
 8. Please note that the scale bar needs to be defined for supplementary figures 3a-b.
 9. Please note that scale bar and its definition are missing for supplementary figures 3c; 4a.
 - Correct order of the manuscript sections to: Author names and affiliations, Abstract, Keywords, Introduction, Results, Discussion, Methods, Acknowledgements, Disclosure and competing interests statement, References, Figure legends, Expanded View Figure legends.
 - Remove "Abbreviation list".
 - Abstract in the manuscript and in our submission system should be the same and should not exceed 175 words. I have gone through your text and revised it (see below). Please review it and amend as you see fit:

Hypoxic-ischemic encephalopathy (HIE) arises from diminished blood flow and oxygen to the neonatal brain during labor, leading to infant mortality or severe brain damage, with a global incidence of 1.5 per 1000 live births. Glucagon-like Peptide 1 Receptor (GLP1-R) agonists, used in type 2 diabetes treatment, exhibit neuroprotective effects in various brain injury models, including HIE. In this study, we observed enhanced neurological outcomes in post-natal day 10 mice with surgically induced hypoxic-ischemic (HI) brain injury after immediate systemic administration of exendin-4 or semaglutide. Short- and long-term assessments revealed improved neuropathology, survival rates, and locomotor function. We explored the mechanisms by which GLP1-R agonists trigger neuroprotection and reduce inflammation following oxygen-glucose deprivation and HI in neonatal mice, highlighting the upregulation of the PI3/AKT signaling pathway and increased cAMP levels. These findings shed light on the neuroprotective and anti-inflammatory effects of GLP1-R agonists in HIE, potentially extending to other neurological conditions, supporting their potential clinical use in treating infants with HIE.

- Add callouts for Fig 2l.
- Rename "Competing interests" to "Disclosure and competing interests statement". We updated our journal's competing interests policy in January 2022 and request authors to consider both actual and perceived competing interests. Please review the policy <https://www.embopress.org/competing-interests> and update your competing interests if necessary.
- Rename "Funding" to "Acknowledgements".
- Please correct the reference citation in the text and reference list. In the text, a reference should be cited by author and year of publication. Include a space between a word and the opening parenthesis of the reference that follows. In the reference list, citations should be listed in alphabetical order. Where there are more than 10 authors on a paper, 10 will be listed, followed by "et al.". Also, remove DOI from references. Please check "Author Guidelines" for more information.
<https://www.embopress.org/page/journal/17574684/authorguide#referencesformat>
- 4) Tables: Please compile the tables in "Appendix" and upload it as a single PDF file, with a table of content including page

numbers. Rename tables to "Appendix Table S1" etc. and update their callouts in the main manuscript text.

5) Funding: Please make sure that information about all sources of funding are complete in both our submission system and in the manuscript.

6) The Paper Explained: Please provide "The Paper Explained" and add it to the main manuscript text. Please check "Author Guidelines" for more information. <https://www.embopress.org/page/journal/17574684/authorguide#researcharticleguide>

7) Synopsis: Every published paper now includes a 'Synopsis' to further enhance discoverability. Synopses are displayed on the journal webpage and are freely accessible to all readers. They include separate synopsis image and synopsis text.

- Synopsis image: Please provide a striking image or visual abstract as a high-resolution jpeg file 550 px-wide x (250-400)-px high to illustrate your article.

- Synopsis text: Please provide a short standfirst (maximum of 300 characters, including space) as well as 2-5 one sentence bullet points that summarise the paper as a .doc file. Please write the bullet points to summarise the key NEW findings. They should be designed to be complementary to the abstract - i.e. not repeat the same text. We encourage inclusion of key acronyms and quantitative information (maximum of 30 words / bullet point). Please use the passive voice.

8) For more information: This space should be used to list relevant web links for further consultation by our readers. Could you identify some relevant ones and provide such information as well? Some examples are patient associations, relevant databases, OMIM/proteins/genes links, author's websites, etc...

9) As part of the EMBO Publications transparent editorial process initiative (see our Editorial at <http://embomolmed.embopress.org/content/2/9/329>), EMBO Molecular Medicine will publish online a Review Process File (RPF) to accompany accepted manuscripts. This file will be published in conjunction with your paper and will include the anonymous referee reports, your point-by-point response and all pertinent correspondence relating to the manuscript. Let us know whether you agree with the publication of the RPF and as here, if you want to remove or not any figures from it prior to publication. Please note that the Authors checklist will be published at the end of the RPF.

10) Please provide a point-by-point letter INCLUDING my comments as well as the reviewer's reports and your detailed responses (as Word file).

I look forward to reading a new revised version of your manuscript as soon as possible.

Yours sincerely,

Zeljko Durdevic

*** Instructions to submit your revised manuscript ***

1) a .docx formatted version of the manuscript text (including Figure legends and tables)

2) Separate figure files*

3) supplemental information as Expanded View and/or Appendix. Please carefully check the authors guidelines for formatting

Expanded view and Appendix figures and tables at
<https://www.embopress.org/page/journal/17574684/authorguide#expandedview>

4) a letter INCLUDING the reviewer's reports and your detailed responses to their comments (as Word file).

5) The paper explained: EMBO Molecular Medicine articles are accompanied by a summary of the articles to emphasize the major findings in the paper and their medical implications for the non-specialist reader. Please provide a draft summary of your article highlighting

6) For more information: There is space at the end of each article to list relevant web links for further consultation by our readers. Could you identify some relevant ones and provide such information as well? Some examples are patient associations, relevant databases, OMIM/proteins/genes links, author's websites, etc...

7) Author contributions: the contribution of every author must be detailed in a separate section.

8) EMBO Molecular Medicine now requires a complete author checklist (<https://www.embopress.org/page/journal/17574684/authorguide>) to be submitted with all revised manuscripts. Please use the checklist as guideline for the sort of information we need WITHIN the manuscript. The checklist should only be filled with page numbers where the information can be found. This is particularly important for animal reporting, antibody dilutions (missing) and exact values and n that should be indicated instead of a range.

9) Every published paper now includes a 'Synopsis' to further enhance discoverability. Synopses are displayed on the journal webpage and are freely accessible to all readers. They include a short stand first (maximum of 300 characters, including space) as well as 2-5 one sentence bullet points that summarise the paper. Please write the bullet points to summarise the key NEW findings. They should be designed to be complementary to the abstract - i.e. not repeat the same text. We encourage inclusion of key acronyms and quantitative information (maximum of 30 words / bullet point). Please use the passive voice. Please attach these in a separate file or send them by email, we will incorporate them accordingly.

You are also welcome to suggest a striking image or visual abstract to illustrate your article. If you do please provide a jpeg file 550 px-wide x 300-800px high.

10) A Conflict of Interest statement should be provided in the main text

11) Please note that we now mandate that all corresponding authors list an ORCID digital identifier. This takes <90 seconds to complete. We encourage all authors to supply an ORCID identifier, which will be linked to their name for unambiguous name identification.

Currently, our records indicate that the ORCID for your account is 0000-0003-0044-0949.

Please click the link below to modify this ORCID:
Link Not Available

- Graphs 800-1,200 DPI
- Photos 400-800 DPI
- Colour (only CMYK) 300-400 DPI"

*Additional important information regarding figures and illustrations can be found at
<https://bit.ly/EMBOPressFigurePreparationGuideline>. See also figure legend preparation guidelines:
<https://www.embopress.org/page/journal/17574684/authorguide#figureformat>

**** Reviewer's comments ****

Referee #1 (Comments on Novelty/Model System for Author):

the key limitation of the animal model is that the authors did not measure the pups' temperatures after returning them to the dam.

As stated in my comments to the authors, there is considerable evidence that HI leads hypothermia (Rectal temperature in the first five hours after hypoxia-ischemia critically affects neuropathological outcomes in neonatal rats. Wood T, Hobbs C, Falck M, Brun AC, Løberg EM, Thoresen M. *Pediatr Res.* 2018 Feb;83(2):536-544)

In turn the effect of many drugs was exaggerated by iatrogenic hypothermia (ref Klaher et al)

Without this additional information, the results are difficult to interpret.

Referee #1 (Remarks for Author):

The authors have partially addressed my concerns.

The key remaining issues are:

1. First, treatment was begun immediately after HI.

The authors have revised the text to say: "An important consideration is that in this study the GLP1-R agonists were administered immediately following HI injury. This may not fully reflect what may happen in a clinical scenario."

This is vague and does not address the now well established understanding that we are not able to start any treatment for HIE immediately, and certainly not in a trial.

The authors may wish to rephrase this as:

"An important consideration is that in this study the GLP1-R agonists were administered immediately following HI injury. In clinical practice, this is not practical; for example in the large trials of therapeutic hypothermia, the average delay was over hours and many took significantly longer {reference to meta-analysis of the RCTs}. Thus the present study establishes proof of principle but further studies of the window of opportunity are now essential."

2. Second, and more importantly, no information is provided on temperature control after HI.

In a very long response, the authors have provided no information on temperature control after HI. As was inferred, they provided a controlled environment before and during HI, but after returning the pups to the dam, we have no information on the pup's temperature. The study by T. Woods shows that at that time after HI, the pups consistently become hypothermic and the authors cannot exclude the possibility that these drugs could be associated with an exaggeration of this pattern.

It is unreasonable to claim that it is not possible or inconsistent with the 3Rs to measure rectal temperatures in sentinel animals. The study by Woods shows that it is possible.

Ideally the authors would do further controlled studies to document the changes in temperature in their setting and the impact of treatment.

The absolute minimum revision must be to state:

A major limitation of the present study is that we did not measure the pups' temperatures after returning them to the dam.

Previous studies confirm that after HI, rodent pups consistently become hypothermic (Wood t) and so we cannot rule out the possibility that the apparent treatment effects were exaggerated by drug induced hypothermia (Klahr et al).

Referee #3 (Remarks for Author):

The authors answered the critiques well and it is an improved manuscript.

Referee #1 (Remarks for Author):

The key limitation of the animal model is that the authors did not measure the pups' temperatures after returning them to the dam.

As stated in my comments to the authors, there is considerable evidence that HI leads to hypothermia (Rectal temperature in the first five hours after hypoxia-ischemia critically affects neuropathological outcomes in neonatal rats. Wood T, Hobbs C, Falck M, Brun AC, Løberg EM, Thoresen M. *Pediatr Res*. 2018 Feb;83(2):536-544)

In turn the effect of many drugs was exaggerated by iatrogenic hypothermia (ref Klaher et al) Without this additional information, the results are difficult to interpret.

The authors have partially addressed my concerns.

The key remaining issues are:

1. First, treatment was begun immediately after HI.

The authors have revised the text to say: "An important consideration is that in this study the GLP1-R agonists were administered immediately following HI injury. This may not fully reflect what may happen in a clinical scenario."

This is vague and does not address the now well established understanding that we are not able to start any treatment for HIE immediately, and certainly not in a trial.

The authors may wish to rephrase this as:

"An important consideration is that in this study the GLP1-R agonists were administered immediately following HI injury. In clinical practice, this is not practical; for example in the large trials of therapeutic hypothermia, the average delay was over hours and many took significantly longer {reference to meta-analysis of the RCTs}. Thus the present study establishes proof of principle but further studies of the window of opportunity are now essential."

Authors: We thank the Referee for their constructive feedback. We have now added the Referee's suggested text and citation to the Discussion section on page 16.

Referee: Second, and more importantly, no information is provided on temperature control after HI.

In a very long response, the authors have provided no information on temperature control after HI. As was inferred, they provided a controlled environment before and during HI, but after returning the pups to the dam, we have no information on the pup's temperature. The study by T. Woods shows that at that time after HI, the pups consistently become hypothermic and the authors cannot exclude the possibility that these drugs could be associated with an exaggeration of this pattern.

It is unreasonable to claim that it is not possible or inconsistent with the 3Rs to measure rectal temperatures in sentinel animals. The study by Woods shows that it is possible.

Ideally the authors would do further controlled studies to document the changes in temperature in their setting and the impact of treatment.

The absolute minimum revision must be to state:

A major limitation of the present study is that we did not measure the pups' temperatures after returning them to the dam. Previous studies confirm that after HI, rodent pups consistently become hypothermic (Wood t) and so we cannot rule out the possibility that the apparent treatment effects were exaggerated by drug induced hypothermia (Klahr et al).

Authors: We have now added the Referee's suggested text and citations to the Discussion section on page 16.

7th May 2024

Dear Prof. Rahim,

We are pleased to inform you that your manuscript is accepted for publication and is now being sent to our publisher to be included in the next available issue of EMBO Molecular Medicine.
